# ENTITY-CENTRIC REINFORCEMENT LEARNING FOR OBJECT MANIPULATION FROM PIXELS

**Dan Haramati, Tal Daniel, Aviv Tamar**

Department of Electrical and Computer Engineering, Technion - Israel Institute of Technology

{haramati, taldanielm}@campus.technion.ac.il; avivt@technion.ac.il

## ABSTRACT

Manipulating objects is a hallmark of human intelligence, and an important task in domains such as robotics. In principle, Reinforcement Learning (RL) offers a general approach to learn object manipulation. In practice, however, domains with more than a few objects are difficult for RL agents due to the curse of dimensionality, especially when learning from raw image observations. In this work we propose a structured approach for visual RL that is suitable for representing multiple objects and their interaction, and use it to learn goal-conditioned manipulation of several objects. Key to our method is the ability to handle goals with dependencies between the objects (e.g., moving objects in a certain order). We further relate our architecture to the generalization capability of the trained agent, based on a theoretical result for compositional generalization, and demonstrate agents that learn with 3 objects but generalize to similar tasks with over 10 objects. Videos and code are available on the project website: https://sites.google.com/view/entity-centric-rl

## 1 INTRODUCTION

Deep Reinforcement Learning (RL) has been successfully applied to various domains such as video games (Mnih et al., 2013) and robotic manipulation (Kalashnikov et al., 2018). While some studies focus on developing general agents that can solve a wide range of tasks (Schulman et al., 2017), the difficulty of particular problems has motivated studying agents that incorporate *structure* into the learning algorithm (Mohan et al., 2023). Object manipulation – our focus in this work – is a clear example for the necessity of structure: as the number of degrees of freedom of the system grows exponentially with the number of objects, the curse of dimensionality inhibits standard approaches from learning. Indeed, off-the shelf RL algorithms struggle with learning to manipulate even a modest number of objects (Zhou et al., 2022).

The factored Markov decision process formulation (factored MDP, Guestrin et al. 2003) factorizes the full state of the environment $\mathcal{S}$ into the individual states of each object, or *entity*, in the system $\mathcal{S}_i$: $\mathcal{S} = \mathcal{S}_1 \otimes \mathcal{S}_2 \otimes ... \otimes \mathcal{S}_N$. A key observation is that if the state transition depends only on *a subset* of the entities (e.g., only objects that are physically near by), this structure can be exploited to simplify learning. In the context of deep RL, several studies suggested structured representations based on permutation invariant neural network architectures for the policy and Q-value functions (Li et al., 2020; Zhou et al., 2022). These approaches require access to the true factored state of the system.

For problems such as robotic manipulation with image inputs, however, how to factor the state (images of robot and objects) into individual entities and their attributes (positions, orientation, etc.) is not trivial. Even the knowledge of which attributes exist and are relevant to the task is typically not given in advance. This problem setting therefore calls for a learning-based approach that can pick up the relevant structure from data. As with any learning-based method, the main performance indicator is generalization. In our setting, generalization may be measured with respect to 3 different factors of variation: (1) the states of the objects in the system, (2) different types of objects, and (3) different *number* of objects than in training, known as *compositional generalization* (Lin et al., 2023).

Our main contribution in this work is a goal-conditioned RL framework for multi-object manipulation from pixels. Our approach consists of two components. The first is an unsupervised object-centric image representation (OCR), which extracts entities and their attributes from image data. The second and key component is a Transformer-based architecture for the policy and Q-function neural networks that we name *Entity Interaction Transformer* (EIT). Different from previous work

such as Zadaianchuk et al. (2020), our EIT is structured such that it can easily model not only relations between goal and state entities, but also entity-entity interactions in the current state. This allows us to learn tasks where interactions between objects are important for achieving the goal, for example, moving objects in a particular order.

Furthermore, the EIT does not require explicit matching between entities in different images, and can therefore handle multiple images from different viewpoints seamlessly. As we find out, multi-view perception is crucial to the RL agent for constructing an internal "understanding" of the 3D scene dynamics from 2D observations in a sample efficient manner. Combined with our choice of Deep Latent Particles (DLP, Daniel & Tamar 2022) image representations, we demonstrate what is, to the best of our knowledge, the most accurate object manipulation from pixels involving more than two objects, or with goals that require interactions between objects.

Finally, we investigate the generalization ability of our proposed framework. Starting with a formal definition of compositional generalization in RL, we show that self-attention based Q-value functions have a fundamental capability to generalize, under suitable conditions on the task structure. This result provides a sound basis for our Transformer-based approach. Empirically, we demonstrate that an EIT trained on manipulating up to 3 objects can perform well on tasks with up to 6 objects, and in certain tasks we show generalization to over 10 objects.

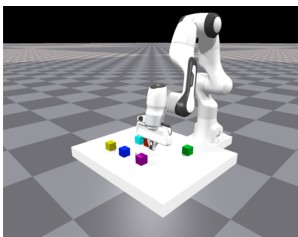 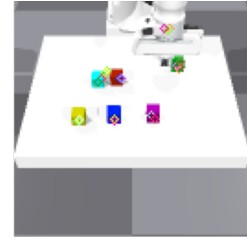 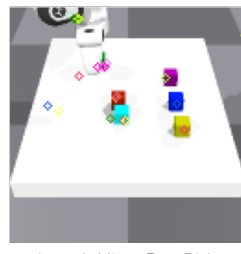

Environment Overview     Agent View 1 - Front     Agent View 2 - Side

Figure 1: *The environment we used for our experiments (left) and how the agent perceives it (middle, right), colored keypoints are the position attribute $z_p$ of particles from the DLP representation.*

## 2 RELATED WORK

**Latent-based Visual RL**: Unstructured latent representations consisting of a *single* latent vector have been widely employed, both in model-free (Levine et al., 2016; Nair et al., 2018; Pong et al., 2019; Yarats et al., 2021) and model-based (Hafner et al., 2023; Mendonca et al., 2021) settings. However, in manipulation tasks involving multiple objects, this approach falls short compared to models based on structured representations (Gmelin et al., 2023; Zadaianchuk et al., 2020).

**Object-centric RL**: Several recent works explored network architectures for a structured representation of the state in model free (Li et al., 2020; Zhou et al., 2022; Mambelli et al., 2022; Zadaianchuk et al., 2022) and model-based (Sanchez-Gonzalez et al., 2018) RL as well as model-based planning methods (Sancaktar et al., 2022). Compared to our approach, the aforementioned methods assume access to ground-truth states, while we learn representations from images.

**Object-centric RL from Pixels**: Several works have explored adopting ideas from state-based structured representation methods to learning from visual inputs. COBRA (Watters et al., 2019), STOVE (Kossen et al., 2019), NCS (Chang et al., 2023), FOCUS (Ferraro et al., 2023), DAFT-RL (Feng & Magliacane, 2023), HOWM (Zhao et al., 2022) and Driess et al. (2023) learn object-centric world models which they use for planning or policy learning to solve multi-object tasks. In contrast to these methods, our method is trained in a model-free setting, makes less assumptions on the problem and is less complex, allowing simple integration with various object-centric representation models and standard online or offline Q-learning algorithms. OCRL (Yoon et al., 2023), Heravi et al. (2023) and van Bergen & Lanillos (2022) have investigated slot-based representations (Greff et al., 2019; Locatello et al., 2020; Singh et al., 2022; Traub et al., 2022) for manipulation tasks in model-free, imitation learning and active inference settings, respectively. The above methods have demonstrated the clear advantages of object-centric representations over non-object-centric alternatives. In this work, we utilize particle-based image representations (Daniel & Tamar, 2022) and extend these findings by tackling more complex manipulation tasks and showcasing generalization capabilities. Closely related to our work, SMORL (Zadaianchuk et al., 2020) employs SCALOR (Jiang et al., 2020), a patch-based image representation for goal-conditioned

manipulation tasks. The key assumption in SMORL is that goal-conditioned multi-object tasks can be addressed sequentially (object by object) and independently, overlooking potential interactions among objects that might influence reaching goals. Our model, on the other hand, considers interaction between entities, leading to improved performance and generalization, as we demonstrate in a thorough comparison with SMORL (see Section 5).

## 3 BACKGROUND

**Goal-Conditioned Reinforcement Learning (GCRL)**: RL considers a Markov Decision Process (MDP, Puterman (2014)) defined as $\mathcal{M} = (\mathcal{S}, \mathcal{A}, P, r, \gamma, \rho_0)$, where $\mathcal{S}$ represents the state space, $\mathcal{A}$ the action space, $P$ the environment transition dynamics, $r$ the reward function, $\gamma$ the discount factor and $\rho_0$ the initial state distribution. GCRL additionally includes a goal space $\mathcal{G}$. The agent seeks to learn a policy $\pi^* : \mathcal{S} \times \mathcal{G} \to \mathcal{A}$ that maximizes the expected return $\mathbb{E}_\pi[\sum_{t=0}^\infty \gamma^t r_t]$, where $r_t = r(s_t, g) : \mathcal{S} \times \mathcal{G} \to \mathbb{R}$ is the immediate reward at time $t$ when the state and goal are $s_t$ and $g$.

**Deep Latent Particles (DLP)**: DLP (Daniel & Tamar, 2022; 2023) is an unsupervised object-centric model for image representation. DLP provides a disentangled latent space structured as a *set* of particles $z = \left\{(z_p, z_s, z_d, z_t, z_f)_i\right\}_{i=0}^{K-1} \in \mathbb{R}^{K \times (6+l)}$, where $K$ is the number of particles, $z_p \in \mathbb{R}^2$ is the position of the particle as $(x, y)$ coordinates in Euclidean pixel space, $z_s \in \mathbb{R}^2$ is a scale attribute containing the $(x, y)$ dimensions of the bounding-box around the particle, $z_d \in \mathbb{R}$ is a pixel space "depth" attribute used to signify which particle is in front of the other in case there is an overlap, $z_t \in \mathbb{R}$ is a transparency attribute and $z_f \in \mathbb{R}^l$ are the latent features that encode the visual appearance of a region surrounding the particle, where $l$ is the dimension of learned visual features. See Appendix A for an extended background.

## 4 METHOD

We propose an approach to solving goal-conditioned multi-object manipulation tasks from images. Our approach is *entity-centric* – it is structured to decompose the input into individual entities, each represented by a latent feature vector, and learn the relationships between them. Our method consists of the following 2 components: (1) **Object-Centric Representation (OCR) of Images** 4.1 – We extract a representation of state and goal images consisting of a set of latent vectors using a pretrained model; (2) **Entity Interaction Transformer (EIT)** 4.2 – We feed the sets of latent vectors, extracted from multiple viewpoints, to a Transformer-based architecture for the RL policy and Q-function neural networks. These two components can be used with standard RL algorithms to optimize a given reward function. We additionally propose a novel image-based reward that is based on the OCR and the Chamfer distance, and corresponds to moving objects to goal configurations. We term this **Chamfer Reward**, and it enables learning entirely from pixels. We begin with a general reasoning that underlies our approach.

**The complexity tradeoff between representation learning and decision making**: An important observation is that the representation learning (OCR) and decision making (EIT) problems are dependent. Consider for example the task of moving objects with a robot arm, as in our experiments. Ideally, the OCR should output the physical state of the robot and each object, which is sufficient for optimal control. However, identifying that the robot is a single entity with several degrees of freedom, while the objects are separate entities, is difficult to learn just from image data, as it pertains to the *dynamics* of the system. The relevant properties of each object can also be task dependent – for example, the color of the objects may only matter if the task's goal depends on it. Alternatively, one may settle for a much leaner OCR component that does not understand the dynamics of the objects nor their relevance to the task, and delegate the learning of this information to the EIT. Our design choice in this paper is the latter, i.e., *a lean OCR and an expressive EIT*. We posit that this design allows to (1) easily acquire an OCR, and (2) handle multiple views and mismatches between the visible objects in the current state and the goal seamlessly. We next detail our design.

### 4.1 OBJECT-CENTRIC REPRESENTATION OF IMAGES

The first step in our method requires extracting a compact disentangled OCR from raw pixel observations. Given a tuple of image observations from $K$ different viewpoints of the state $(I_1^s, \ldots, I_K^s)$ and goal $(I_1^g, \ldots, I_K^g)$, we process each image *separately* using a pretrained Deep Latent Particles (DLP) (Daniel & Tamar, 2022; 2023) model, extracting a set of $M$ vectors $\{p_m^k\}_{m=1}^M$, $k$ indexing the viewpoint of the source image, which we will refer to as (latent) particles. We denote particle

$m$ of state image $I_k^s$ by $p_m^k$ and of goal image $I_k^g$ by $q_m^k$. We emphasize that there is *no alignment* between particles from different views (e.g., $p_m^1, p_m^2$ can correspond to different objects) or between state and goal of the same view (e.g., between $p_m^1$ and $q_m^1$). The vectors $p_m^k, q_m^k \in \mathbb{R}^{6+l}$ contain different attributes detailed in the DLP section of the Background 3. In contrast to previous object-centric approaches that utilize patch-based (Zadaianchuk et al., 2020) or slot-based (Yoon et al., 2023) representations, we adopt DLP, which has recently demonstrated state-of-the-art performance in single-image decomposition and various downstream tasks. The input to the goal-conditioned policy is a tuple of the $2K$ sets: $\left( \{p_m^1\}_{m=1}^M, \ldots, \{p_m^K\}_{m=1}^M, \{q_m^1\}_{m=1}^M \ldots, \{q_m^K\}_{m=1}^M \right)$. For the Q-function, $Q(s, a, g)$, an action particle $p_a \in \mathbb{R}^{6+l}$ is added to the input, obtained by learning a projection from the action dimension $d_a$ to the latent particle dimension.

**Pretraining the DLP**: In this work, we pretrain the DLP from a dataset of images collected by a random policy. We found that in all our experiments, this simple pretraining was sufficient to obtain well performing policies even though the image trajectories in the pretraining data are different from trajectories collected by an optimal policy. We attribute this to the tradeoff described above – the lean single-image based DLP OCR is complemented by a strong EIT that can account for dynamics necessary to solve the task.

### 4.2 Entity-Centric Architecture

We next describe the EIT, which processes the OCR entities into an action or Q-value. As mentioned above, our choice of a lean OCR requires the EIT to account for the dynamics of the entities and their relation to the task. In particular, we have the following desiderata from the EIT: (1) **Permutation Invariance**; (2) **Handle goal-based RL**; (3) **Handle multiple views**; (4) **Compositional generalization**. Proper use of the attention mechanism[1] provides us with (1). The main difficulty in (2) and (3) is that particles in different views and the goal are not necessarily matched. Thus, we designed our EIT to seamlessly handle unmatched entities. For (4), we compose the EIT using Transformer (Vaswani et al., 2017) blocks. As we shall show in Section 4.4, this architecture has a fundamental capacity to generalize. An outline of the architecture is presented in Figure 2. Compared to previous goal-conditioned methods' use of the attention mechanism, we use it to explicitly model relationships between entities from both state and goal across multiple viewpoints and do not assume privileged entity-entity matching information. A more detailed comparison can be found in Appendix E. We now describe the EIT in detail:

**Input** - The EIT policy receives the latent particles extracted from both the current state images and the goal images as input. We inject information on the source viewpoint of each state and goal particle with an additive encoding which is learned concurrently with the rest of the network parameters.
**Forward** - State particles $\left( \{p_j^1\}_{j=1}^M, \ldots, \{p_j^N\}_{j=1}^M \right)$ are processed by a sequence of Transformer blocks: $SA \rightarrow CA \rightarrow SA \rightarrow AA$, denoting *Self-Attention SA*, *Cross-Attention CA*, and *Aggregation-Attention AA*, followed by an MLP.
**Goal-conditioning** - We condition on the goal particles $\left( \{q_k^1\}_{k=1}^M, \ldots, \{q_k^N\}_{k=1}^M \right)$ using the $CA$ block between the state (provide the queries) and goal (provide the keys and values) particles.
**Permutation Invariant Output** - The $AA$ block reduces the set to a single-vector output and is implemented with a $CA$ block between a single particle with learned features (provides the query) and the output particles from the previous block (provide the keys and values). This permutation invariant operation on the processed state particles, preceded by permutation equivariant Transformer blocks, results in an output that is invariant to permutations of the particles in each set. The aggregated particle is input to an MLP, producing the final output (action/value).
**Action Entity** - The EIT Q-function, in addition to the state and goal particles, receives an action as input. The action is projected to the dimension of the particles and added to the input set. Treating the action as an individual entity proved to be a significant design choice, see ablation study in C.4.

### 4.3 Chamfer Reward

We define an image-based reward from the DLP representations of images as the Generalized Density-Aware Chamfer (GDAC) distance between state and goal particles, which we term *Chamfer reward*. The standard Chamfer distance is defined between two sets and measures the average

---

[1]We provide a detailed exposition to attention in Appendix A.3.

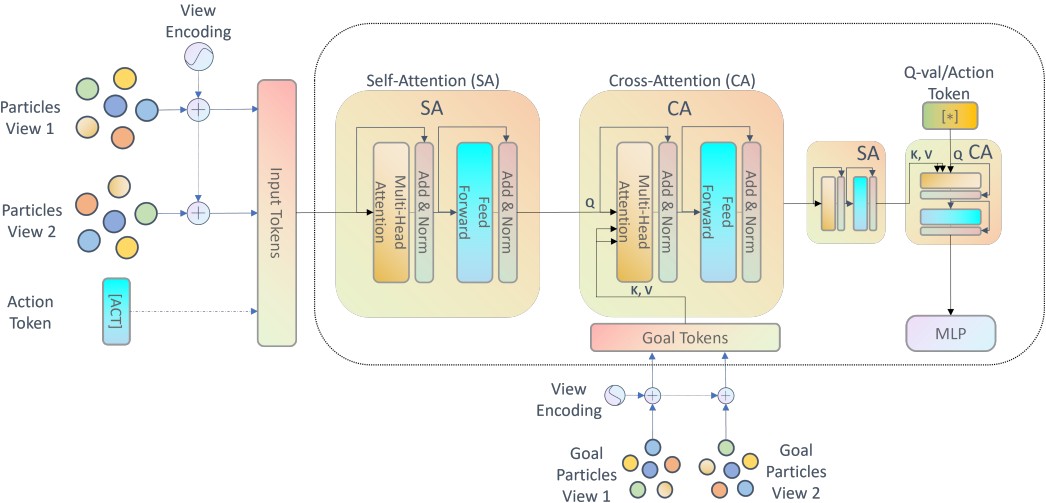

Figure 2: *Outline of the Entity Interaction Transformer (EIT) - Sets of state and goal particles from multiple views with an additive view encoding are input to a sequence of Transformer blocks. For the Q-function, an action particle is added. We condition on goals with cross-attention. Attention-based aggregation reduces the set to a single vector, followed by an MLP that produces the final output.*

distance between each entity in one set to the closest entity in the other. The *Density-Aware* Chamfer distance (Wu et al., 2021) takes into account the fact that multiple entities from one set can be mapped to the same entity in the other set by reweighing their contribution to the overall distance accordingly. The *Generalized* Density-Aware Chamfer distance, decouples the distance function that is used to match between entities and the one used to calculate the distance between them. For space considerations, we elaborate on the Chamfer reward in Appendix B.

### 4.4 COMPOSITIONAL GENERALIZATION

In our context, Compositional Generalization (CG) refers to the ability of a policy to perform a multi-object task with a different number of objects than it was trained on. In this section, we formally define a notion of CG for RL, and show that under sufficient conditions, a self-attention based Q-function architecture can obtain it. This result motivates our choice of the Transformer-based EIT, and we shall investigate it further in our experiments. We give our main result here, and provide an in-depth discussion and full proofs in Appendix F.

Let $\tilde{\mathcal{S}}$ denote the state space of a single object, and consider an $N$-object MDP as an MDP with a factored state space $\mathcal{S}^N = \mathcal{S}_1 \otimes \mathcal{S}_2 \otimes ... \otimes \mathcal{S}_N, \forall i : \mathcal{S}_i \in \tilde{\mathcal{S}}$, action space $\mathcal{A}$, reward function $r$ and discount factor $\gamma$. Without loss of generality, assume that $0 \le r \le R_{max} = 1$.

Naturally, tasks which are completely different for every $N$ would not be amenable for CG. We constrain the set of tasks by assuming a certain structure of the optimal Q-function. We say that *a class of functions admits compositional generalization* if, after training a Q-function from this class on $M$ objects, the test error on $M + k$ objects grows at most linearly in $k$. The more expressive the function class that admits CG is, the greater the chance that it would apply for the task at hand. In the following, we prove our main result – CG for the class of self-attention functions. This class is suitable for tasks where the optimal policy involves a similar procedure that must be applied to each object, such as the tasks we consider in Section 5.

**Assumption 1.** $\forall S \in \mathcal{S}^N, \forall a \in \mathcal{A}, \forall N \in \mathbb{N}$ we have:
$Q^* (s_1, ..., s_N, a) = \frac{1}{N} \sum_{i=1}^{N} \tilde{Q}_i^* (s_1, ..., s_N, a)$, where
$\tilde{Q}_i^* (s_1, ..., s_N, a) = \frac{1}{\sum_{j=1}^{N} \alpha^* (s_i, s_j, a)} \sum_{j=1}^{N} \alpha^* (s_i, s_j, a) v^* (s_j, a), \ \alpha^* (\cdot) \in \mathbb{R}^+.$

The following result shows that when obtaining an $\varepsilon$-optimal Q-function for up to $M$ objects where the attention weights are $\delta$-optimal, the sub-optimality w.r.t. $M + k$ objects grows at most linearly in $k$.

**Theorem 2.** *Let Assumption 1 hold. Let $\hat{Q}$ be an approximation of $Q^*$ with the same structure. Assume that $\forall s \in \mathcal{S}^N$, $\forall a \in \mathcal{A}$, $\forall N \in [1, M]$ we have $\left| \hat{Q}\left(s_1, ..., s_N, a\right) - Q^*\left(s_1, ..., s_N, a\right) \right| < \varepsilon$, and $\left| \frac{\alpha(s_i, s_j, a)}{\sum_{j=1}^N \alpha(s_i, s_j, a)} - \frac{\alpha^*(s_i, s_j, a)}{\sum_{j=1}^N \alpha^*(s_i, s_j, a)} \right| < \delta$. Then, $\forall s \in \mathcal{S}^{M+k}$, $\forall a \in \mathcal{A}$, $\forall k \in [1, M-1]$:*

$$\left| \hat{Q}\left(s_1, ..., s_{M+k}, a\right) - Q^*\left(s_1, ..., s_{M+k}, a\right) \right| \leq 3\varepsilon + \frac{3(M+k) + 2}{1 - \gamma}\delta.$$

### 4.5 TRAINING AND IMPLEMENTATION DETAILS

We developed our method with the off-policy algorithm TD3 (Fujimoto et al., 2018) along with hindsight experience replay (HER, Andrychowicz et al. 2017). In principal, our approach is not limited to actor-critic algorithms or to the online setting, and can be used with any deep Q-learning algorithm, online or offline. We pre-train a single DLP model on images from multiple viewpoints of rollouts collected with a random policy. We convert state and goal images to object-centric latent representations with the DLP encoder before inserting them to the replay buffer. We use our EIT architecture for all policy and Q-function neural networks. Further details and hyper-parameters can be found in Appendix D. Our code is publicly available on `https://github.com/DanHrmti/ECRL`.

## 5 EXPERIMENTS

We design our experimental setup to address the following aspects: (1) benchmarking our method on multi-object manipulation tasks from pixels; (2) assessing the significance of accounting for interactions between entities for the RL agent's performance; (3) evaluating the scalability of our approach to increasing number of objects; (4) analyzing the generalization capabilities of our method.

**Environments** We evaluate our method on several simulated tabletop robotic object manipulation environments implemented with IsaacGym (Makoviychuk et al., 2021). The environment includes a robotic arm set in front of a table with a varying number of cubes in different colors. The agent observes the state of the system through a number of cameras in fixed locations, and performs actions in the form of deltas in the end effector coordinates $a = (\Delta x_{ee}, \Delta y_{ee}, \Delta z_{ee})$. At the beginning of each episode, the cube positions are randomly initialized on the table, and a goal configuration is sampled similarly. The goal of the agent is to push the cubes to match the goal configuration. We categorize a suite of tasks as follows (see Figure 3):

`N-Cubes`: Push $N$ different-colored cubes to their goal location.
`Adjacent-Goals`: A `3-Cubes` setting where goals are sampled randomly on the table such that all cubes are adjacent. This task requires accounting for interactions between objects.
`Small-Table`: A `3-Cubes` setting where the table is substantially smaller. This task requires to accurately account for all objects in the scene at all times, to avoid pushing blocks off the table.
`Ordered-Push`: A `2-Cubes` setting where a narrow corridor is set on top of the table such that its width can only fit a single cube. We consider two possible goal configurations: red cube in the rear of the corridor and green cube in the front, or vice versa. This task requires to fulfill the goals in a certain order, otherwise the agent fails (pulling a block out of the corridor is not possible).
`Push-2T`: Push 2 T-shaped blocks to a single goal *orientation*.

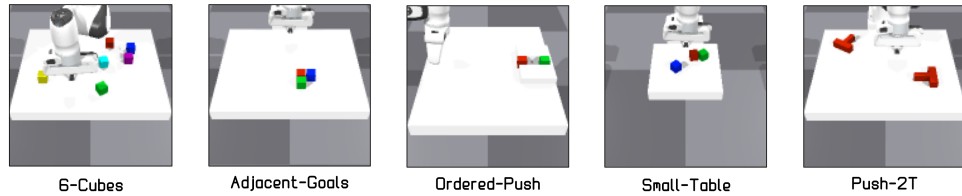

Figure 3: *The simulated environments used for experiments in this work.*

**Reward** The reward calculated from the ground-truth state of the system, which we refer to as the ground-truth (GT) reward, is the mean negative $L_2$ distance between each cube and its desired goal position on the table. The image-based reward calculated from the DLP OCR for our method is the negative GDAC distance (see Eq. 1) between state and goal sets of particles, averaged over viewpoints. Further reward details can be found in the Appendix D.1.

**Evaluation Metrics** We evaluate the performance of the agents on several metrics. In this environment, we define an episode a *success* if all $N$ objects are at a threshold distance from their desired

goal. The metric most closely captures task success, but does not capture intermediate success or timestep efficiency. To this end, we additionally evaluate based on *success fraction*, *maximum object distance*, *average object distance* and *average return*. For a formal definition of these metrics see Appendix D.1. All results show means and standard deviations across 3 random seeds.

**Baselines** We compare our method with the following baselines:
`Unstructured` – A single-vector latent representation of images using a pre-trained VAE is extracted from multiple viewpoints from both state and goal images and then concatenated and fed to an MLP architecture for the policy and Q-function neural networks. This baseline corresponds to methods such as Nair et al. (2018), with the additional multi-view data as in our method.
`SMORL` – We re-implement SMORL (Zadaianchuk et al., 2020), extending it to multiview inputs and tune its hyper-parameters for the environments in this work. Re-implementation details are available in Appendix D.4. We use DLP as the pre-trained OCR for this method for a fair comparison. Note that image-based SMORL cannot utilize GT reward since it requires matching the particle selected from the goal image at the beginning of each training episode to the corresponding object in the environment. We therefore do not present such experiments.

**Object-centric Pretraining** All image-based methods in this work utilize pre-trained unsupervised image representations trained on data collected with a random policy. For the object-centric methods, we train a single DLP model on data collected from the `6-Cubes` environment. We found it generalizes well to fewer objects and changing backgrounds (e.g., smaller table). For the `Push-2T` environment we trained DLP on data collected from `Push-T` (single block) which generalized well to 2 blocks. Figure 14 illustrates the DLP decomposition of a single training image. For the non-OCR baselines, we use a mixture of data from the `1/2/3-Cubes` environments to learn a latent representation with a $\beta$-VAE (Higgins et al., 2017). More information on the architectures and hyper-parameters is available in Appendix D.3.

**Experiment Outline** We separate our investigation into 3 parts. In the first part, we focus on the design of our OCR and EIT, and how it handles complex interactions between objects. We study this using the GT reward, to concentrate on the representation learning question. When comparing with baselines, we experiment with both the OCR and a ground-truth state representation[2]. To prove that our method indeed handles complex interactions, we shall show that our method with the OCR outperforms baselines with GT state on complex tasks. In the second part, we study compositional generalization. In this case we also use the GT reward, with similar motivation as above. Finally, in the third part we evaluate our method using the Chamfer reward. The Chamfer reward is calculated by filtering out particles that do not correspond to objects of interest such as the agent. Due to lack of space this part is detailed in Appendix B.1. An ablation study of key design choices such as incorporating multi-view inputs can be found in Appendix C.4.

## 5.1 MULTI-OBJECT MANIPULATION

We evaluate the different methods with GT rewards on the environments detailed above. Results are presented in Figure 4 and Table 1. We observe that with a single object, all methods succeed, yet the unstructured baselines are less sample efficient. With more than 1 object, the image-based unstructured baseline is not able to learn at all, while the unstructured state-based baseline is significantly outperformed by the structured methods. Our method and SMORL reach similar performance in the state-based setting, SMORL being more sample efficient. This is expected as SMORL essentially learns single object manipulation, regardless of the number of cubes in the environment. Notably, on `3-Cubes`, *our image-based method surpasses the unstructured state-based method*.

In environments that require interaction between objects – `Adjacent-Goals`, `Small-Table` and `Ordered-Push` – our method outperforms SMORL using state input. Moreover, with image inputs our method outperforms SMORL with state inputs (significantly on `Ordered-Push`, yet marginally on `Small-Table`, `Adjacent-Goals`), demonstrating that SMORL is fundamentally limited in performing these more complex tasks.

We present preliminary results on the `Push-2T` environment. For a visualization of the task and performance results see Figure 5. The success in this task demonstrates the following additional capabilities of our proposed method: (1) Handling objects that have more complex physical properties

---

[2]The ground truth state is $s_i = (x_i, y_i)$, $g_i = (x_i^g, y_i^g)$ the $(x, y)$ coordinates of the state and goal of entity $i$ respectively. We detail how this state is input to the networks in Appendix D.1

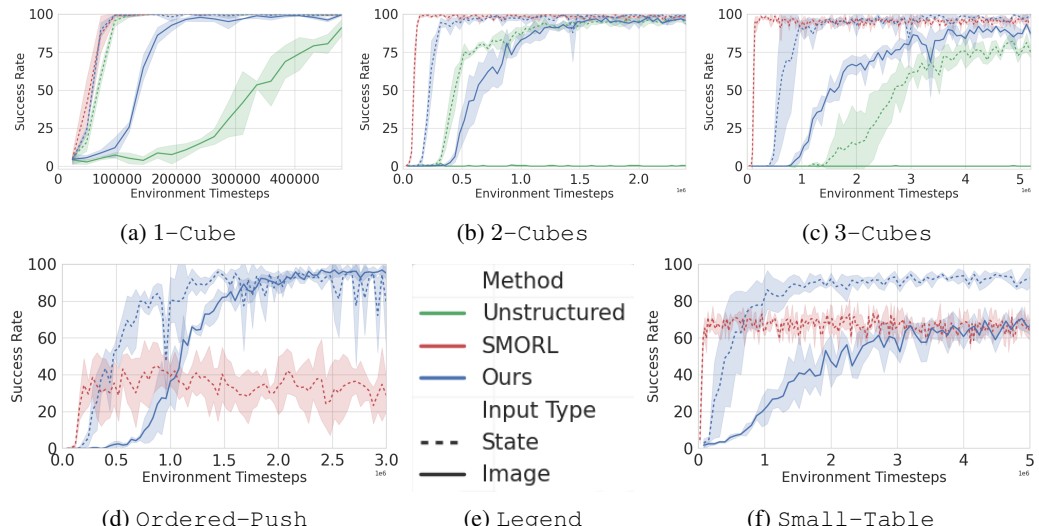

(a) `1-Cube`  (b) `2-Cubes`  (c) `3-Cubes`

(d) `Ordered-Push`  (e) Legend  (f) `Small-Table`

Figure 4: ***Success Rate vs. Environment Timesteps*** – *Values calculated on* 96 *randomly sampled goals. Methods with input type 'State' are presented in dashed lines and learn from GT state observations, otherwise, from images. Our method performs better than or equivalently to the best performing baseline in each category (state/image-based). In the environments requiring object interaction ((d), (f)), our method achieves significantly better performance than SMORL. Notably, our image-based method matches/surpasses state-based SMORL.*

| Method | Success Rate | Success Fraction | Max Obj Dist | Avg Obj Dist | Avg Return |
|---|---|---|---|---|---|
| Ours (State) | $0.963 \pm 0.005$ | $0.982 \pm 0.005$ | $0.022 \pm 0.002$ | $0.014 \pm 0.002$ | $-0.140 \pm 0.008$ |
| SMORL (State) | $0.716 \pm 0.006$ | $0.863 \pm 0.005$ | $0.063 \pm 0.003$ | $0.031 \pm 0.001$ | $-0.233 \pm 0.004$ |
| Ours (Image) | $0.710 \pm 0.016$ | $0.883 \pm 0.005$ | $0.044 \pm 0.003$ | $0.027 \pm 0.001$ | $-0.202 \pm 0.007$ |

Table 1: ***Performance Metrics for*** `Adjacent-Goals` – *Methods trained on* 3`-Cubes` *and evaluated on* `Adjacent-Goals`. *Values calculated on* 400 *random goals per random seed.*

which affect dynamics. (2) The ability of the EIT to infer object **properties that are not explicit in the latent representation** (i.e. inferring orientation from latent particle visual attributes), and accurately manipulate them in order to achieve desired goals.

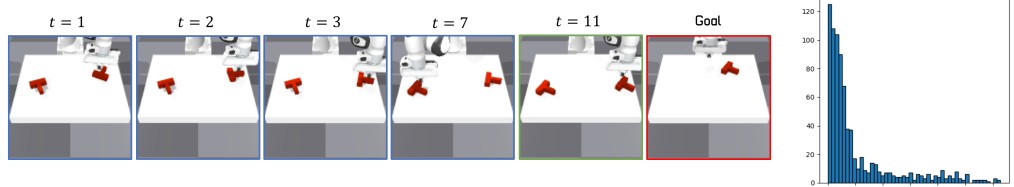

Figure 5: ***Left*** – *Rollout of an agent trained on the* `Push-2T` *task.* ***Right*** – *Distribution of object angle difference (radians) from goal. Values of* 400 *episodes with randomly initialized goal and initial configurations.*

## 5.2 COMPOSITIONAL GENERALIZATION

In this section, we investigate our method's ability to achieve zero-shot compositional generalization. Agents were trained *from images* with our method using GT rewards and require purely image inputs during inference. We present several inference scenarios requiring compositional generalization:

**Different Number of Cubes than in Training** - We train an agent on the 3`-Cubes` environment and deploy the obtained policy on the $N$`-Cubes` environment for $N \in [1, 6]$. Colors are sampled uniformly out of 6 options and are distinct for each cube. Visual results on 6 cubes are presented in Figure 6 (left) and evaluation metrics in Table 4 (Appendix). We see that our agent generalizes

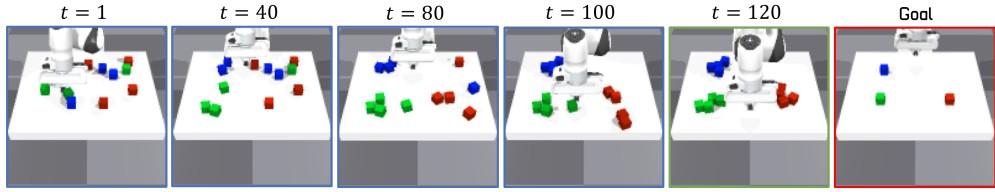

Figure 6: **Left** – *Rollout of an agent trained on* 3 *cubes generalizing to* 6 *cubes.* **Right** – *The average return of an agent trained on* 3 *cubes vs. the number of cubes in the environment it was deployed in during inference. Values are averaged over* 400 *episodes with randomly initialized goal and initial configurations. Note that the graph is approximately linear, corresponding with Theorem 2.*

to a changing number of objects with some decay in performance as the number of objects grows. Notably, the decay in the average return, plotted in Figure 6 (right), is approximately linear in the number of cubes, which corresponds with Theorem 2.

**Cube Sorting** - We train an agent on the `3-Cubes` environment with constant cube colors (red, green, blue). During inference, we provide a goal image containing $X \leq 3$ cubes of different colors and then deploy the policy on an environment containing $4X$ cubes, 4 of each color. The agent sorts the cubes around each goal cube position with matching color, and is also able to perform the task with cube colors unseen during RL training. Visual results on 12 cubes in 3 colors are presented in Figure 7. We find these results exceptional, as they require compositional generalization from both the EIT policy (trained on 3 cubes) and the DLP model (trained on 6 cubes) to significantly more cubes, occupying a large portion of the table's surface.

Figure 7: *Rollout of an agent trained on* 3 *cubes, then provided a goal image containing* 3 *different colored cubes and deployed in an environment with* 12 *cubes,* 4 *of each color. The agent sorts the cubes around each goal position with matching color.*

Further results and analysis of our agent's generalization capabilities such as generalizing to cube properties not seen during training are detailed in Appendix C. Videos are available on our website.

## 6 CONCLUSION AND FUTURE WORK

In this work we proposed an RL framework for object manipulation from images, composed of an off-the-shelf object-centric image representation, and a novel Transformer-based policy architecture that can account for multiple viewpoints and interactions between entities. We have also investigated compositional generalization, starting with a formal motivation, and concluding with experiments that demonstrate non-trivial generalization behavior of our trained policies.

**Limitations:** Our approach requires a pretrained DLP. In this work, our DLP was pretrained from data collected using a random policy, which worked well on all of our domains. However, more complex tasks may require more sophisticated pretraining, or an online approach that mixes training the DLP with the policy. Our results with Chamfer rewards show worse performance than with ground truth reward. This hints that directly running our method on real robots may be more difficult than in simulation due to the challenges of reward design.

**Future work:** One interesting direction is understanding what environments are solvable using our approach. While the environments we investigated here are more complex than in previous studies, environments with more complex interactions between objects, such as with interlocking objects or articulated objects, may be difficult to solve due to exploration challenges. Other interesting directions include multi-modal goal specification (e.g., language), and more expressive sensing (depth cameras, force sensors, etc.), which could be integrated as additional input entities to the EIT.

## 7 REPRODUCIBILITY STATEMENT

We are committed to ensuring the reproducibility of our work and have taken several measures to facilitate this. Appendix D contains detailed implementation notes and hyper-parameters used in our experiments. Furthermore, we make our code openly available to the community to facilitate further research in the following repository: `https://github.com/DanHrmti/ECRL`. The codebase includes the implementation of the environments and our proposed method, as well as scripts for reproducing the experiments reported in this paper. In addition, Appendix F contains in depth details of the theoretical portion of this paper which include the full proof for Theorem 2.

## 8 ACKNOWLEDGMENTS AND DISCLOSURE OF FUNDING

This research was Funded by the European Union (ERC, Bayes-RL, 101041250). Views and opinions expressed are however those of the author(s) only and do not necessarily reflect those of the European Union or the European Research Council Executive Agency (ERCEA). Neither the European Union nor the granting authority can be held responsible for them.

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

# A  EXTENDED BACKGROUND

## A.1  DEEP Q-LEARNING

Goal-conditioned Deep Q-learning approaches learn a Q-function $Q^\pi(s, a, g) : \mathcal{S} \times \mathcal{A} \times \mathcal{G} \rightarrow \mathbb{R}$ parameterized by a deep neural network $Q_\theta(s, a, g)$ with parameters $\theta$, which approximates the expected return given goal $g$ when taking action $a$ at state $s$ and then following the policy $\pi$. $Q_\theta(s, a, g)$ is learned via minimization of the temporal difference (TD, Sutton (1988)) objective given a state, action, next state, goal tuple $(s, a, s', g)$: $\mathcal{L}_{TD}(\theta) = [r(s, g) + \gamma Q_{\bar\theta}(s', \pi(s', g), g) - Q_\theta(s, a, g)]^2$, where $Q_{\bar\theta}(s', a', g)$ is a target network with parameters $\bar\theta$ which are constant under the TD objective. Specifically in off-policy actor-critic algorithms (Fujimoto et al., 2018), a policy network $\pi_\phi(s, g)$ (actor) with parameters $\phi$ is learned concurrently with the Q-function network (critic) with the objective of maximizing it with respect to the action: $\mathcal{L}_\pi(\phi) = -Q_\theta(s, \pi_\phi(s, g), g)$. Alternating between data collection via deploying $\pi_\phi$ in the environment and updating $Q_\theta, \pi_\phi$ with this data is expected to converge to an approximately optimal Q-function $Q_\theta \approx Q^*$ and policy $\pi_\phi \approx \pi^*$.

## A.2  DEEP LATENT PARTICLES (DLP)

DLP (Daniel & Tamar, 2022) is a VAE-based unsupervised object-centric model for images. The key idea in DLP is that the latent space of the VAE is structured as a set of $K$ particles $z = [z_f, z_p] \in \mathbb{R}^{K \times (l+2)}$, where $z_f \in \mathbb{R}^{K \times l}$ is a latent feature vector that encodes the visual appearance of each particle, and $z_p \in \mathbb{R}^{K \times 2}$ encodes the position of each particle as $(x, y)$ coordinates in Euclidean pixel-space. This requires several structural modifications to the standard VAE as described below.

**Prior Modeling:** The prior $p(z|x)$ in DLP is conditioned on the image $x$, and has a different structure for $z_f$ and $z_p$. $p(z_f|x) = p(z_f)$ is modeled by a set of standard zero-mean Gaussians. $p(z_p|x)$ consists of Gaussians centered on a set of keypoint proposals which are produced by a CNN applied to individual patches of the image, followed by a *spatial-softmax* (SSM, Jakab et al. 2018; Finn et al. 2016).

**Encoder:** DLP employs a CNN-based encoder that maps the image to a set of means and log-variances for the keypoint positions $z_p$. The appearance features $z_f$ are encoded from a region around each keypoint (termed *glimpse*) using a Spatial Transformer Network (Jaderberg et al. 2015).

**KL Loss Term:** As the posterior keypoints $S_1$ and the prior keypoint proposals $S_2$ are *unordered sets* of Gaussian distributions, the KL term for the position latents is replaced with the Chamfer-KL: $d_{CH-KL}(S_1, S_2) = \sum_{z_p \in S_1} \min_{z'_p \in S_2} KL(z_p \| z'_p) + \sum_{z'_p \in S_2} \min_{z_p \in S_1} KL(z_p \| z'_p)$.

**Decoder:** Each particle is decoded separately to reconstruct its glimpse RGBA patch. The glimpses are then composed with respect to their encoded positions to stitch the final image.

All components of the DLP model are learned end-to-end in an unsupervised fashion, by maximizing the ELBO (i.e., minimizing the reconstruction loss and the (Chamfer) KL-divergence between the posterior and prior distributions).

**DLPv2:** Daniel & Tamar (2023) expands upon the original DLP's definition of a latent particle, as described above, by incorporating additional attributes. DLPv2 provides a disentangled latent space structured as a *set* of $K$ foreground particles $z = \{(z_p, z_s, z_d, z_t, z_f)_i\}_{i=0}^{K-1} \in \mathbb{R}^{K \times (6+l)}$. $z_p \in \mathbb{R}^2$ and $z_f \in \mathbb{R}^l$ remain unchanged. $z_s \in \mathbb{R}^2$ is a scale attribute containing the $(x, y)$ dimensions of the bounding-box around the particle, $z_d \in \mathbb{R}$ is a pixel space "depth" attribute used to signify which particle is in front of the other in case there is an overlap and $z_t \in \mathbb{R}$ is a transparency attribute. Moreover, it assigns a single abstract particle for the background that is always located in the center of the image and described only by $m_{\text{bg}}$ latent background visual features, $z_{\text{bg}} \sim \mathcal{N}(\mu_{\text{bg}}, \sigma_{\text{bg}}^2) \in \mathbb{R}^{m_{\text{bg}}}$. **We discard the background particle from the latent representation after pre-training the DLP for RL purposes**. Training of DLPv2 is similar to the standard DLP with modifications to the encoding and decoding that take into account the finer control over inference and generation due to the additional attributes.

### A.3 THE ATTENTION MECHANISM

Attention (Bahdanau et al., 2015; Vaswani et al., 2017) denoted $A(\cdot, \cdot)$ is an operator between two sets of vectors, $X = \{x_i\}_{i=1}^N$ and $Y = \{y_j\}_{j=1}^M$, producing a third set of vectors $Z = \{z_i\}_{i=1}^N$. For simplicity, we describe the case were all input, output and intermediate vectors are in $\mathbb{R}^d$. Denote the key, query and value projection functions $q(\cdot),\ k(\cdot),\ v(\cdot) : \mathbb{R}^d \to \mathbb{R}^d$ respectively. The attention operator is defined as $A(X, Y) = Z$ where:

$$z_i = \sum_{j=1}^N \alpha\left(x_i, y_j\right) v\left(y_j\right), \ \ \alpha\left(x_i, y_j\right) = \mathrm{softmax}_j \left(\frac{q\left(x_i\right) \cdot k\left(y_j\right)}{\sqrt{d}}\right) \in \mathbb{R}.$$

Namely, each element of the output set $Z$ is a weighted average of the projected input set $Y$: $\{v(y_j)\}_{j=1}^M$, where the *attention weights* $\alpha_{ij} = \alpha\left(x_i, y_j\right)$ express the "relevance" of $y_j$ for computing output $z_i$ corresponding to $x_i$. An important property of $A(X, Y)$ is that it is *equivariant* to permutations of $X$ (permutation of elements in $X$ results in the same permutation in the output elements in $Z$, with no change in individual elements' values) and *invariant* to permutations of $Y$ (permutation of elements in $Y$ does not change the output $Z$). In the special case where $X = Y$, the operation is termed *self-attention* (SA), and otherwise, *cross-attention* (CA).

## B  CHAMFER REWARD

We desire a reward that captures the task of moving objects to goal configurations. However, because particles in different images are not aligned, and some particles may be occluded or missing, we cannot directly construct a reward based on distances between the particles. Instead, we define a reward from the DLP representations of images as the Generalized Density-Aware Chamfer (GDAC) distance between state and goal particles, which we term *Chamfer reward*. The GDAC distance is defined between two sets $X = \{x_i\}_{i=1}^N$, $Y = \{y_j\}_{j=1}^M$, $x_i, y_i \in \mathbb{R}^d$ in the following manner:

$$Dist_{GDAC}\left(X, Y\right) = \frac{1}{\sum_j I(|X_j| > 0)} \sum_j \frac{1}{|X_j| + \varepsilon} \sum_{x \in X_j} D_1(x, y_j) + \frac{1}{\sum_i I(|Y_i| > 0)} \sum_i \frac{1}{|Y_i| + \varepsilon} \sum_{y \in Y_i} D_1(y, x_i) \tag{1}$$

where $X_j = \left\{x_i | \arg\min_{y_k \in Y}\left(D_2\left(x, y_k\right)\right) = j\right\}$, $Y_i = \left\{y_j | \arg\min_{x_k \in X}\left(D_2\left(y, x_k\right)\right) = i\right\}$ $D_1(x, y)$ and $D_2(x, y)$ are two distance functions between entities. The standard Chamfer distance is obtained by setting $D_1(x, y) = D_2(x, y) = \|x - y\|_2^2$ and substituting $\frac{1}{\sum_j I(|X_j| > 0)} \cdot \frac{1}{|X_j| + \varepsilon}$, $\frac{1}{\sum_i I(|Y_i| > 0)} \cdot \frac{1}{|Y_i| + \varepsilon}$ with $\frac{1}{|X|}$, $\frac{1}{|Y|}$ respectively, removing the inner sums in both terms.

The Chamfer distance measures the average distance between each entity in $X$ and the closest entity to it in $Y$ and vice versa. The *Density-Aware* Chamfer distance (Wu et al., 2021) takes into account the fact that multiple entities from one set can be mapped to the same entity in the other set, and re-weights their contribution to the overall distance accordingly. The *Generalized* Density-Aware Chamfer distance, decouples the distance function that is used to match between entities $D_2(x, y)$ and the one used to calculate the distance between them $D_1(x, y)$. Decoupling these two allows using entity-identifying attributes for matching while calculating the actual distances between matching entities based on localization features. For example, we can use the DLP visual features $z_f$ to match between objects in the current and goal images, and then measure their distance using the $(x, y)$ coordinate attributes $z_p$.

### B.1  FOCUSED CHAMFER REWARD

In many robotic object manipulation settings, we do not care about the robot in our goal specification as long as the objects reach the desired configuration. In order to consider only a subset of the entities for the image-based reward (e.g. particles corresponding to objects and not the agent), we train a simple multi-layer perceptron (MLP) binary classifier on the latent visual features of the DLP representation, differentiating objects of interest from the rest of the particles. We train this classifier on annotated particles extracted from 20 images of the environment. Annotation required 5 minutes of our time and training the classifier itself took just a few seconds. We then filter out particles based on the classifier output before inputting them to the Chamfer reward. We emphasize

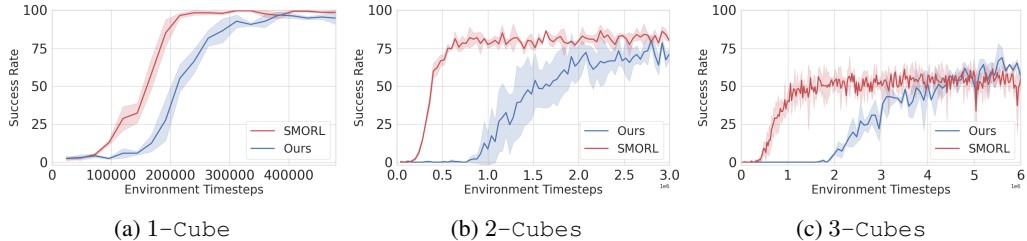

|  |  |  |
|---|---|---|
| (a) 1-Cube | (b) 2-Cubes | (c) 3-Cubes |

Figure 8: *Success Rate vs. Environment Timesteps (Image-Based Rewards)* – *Values calculated based on* 96 *randomly sampled goals.*

that this supervision is only required for training the classifier which is used for the image-based reward exclusively during RL training, and is very simple to acquire both in simulation and the real world.

## B.2 TRAINING WITH THE CHAMFER REWARD

We compare our method to SMORL, trained entirely from images. Results on $N$-Cubes for $N \in \{1, 2, 3\}$ are presented in Figure 8 and Table 2. Training our method with the image-based reward obtains lower success rates compared to training with the GT reward. While this is expected, we believe the large differences are due to noise originated in the DLP representation and occlusions, which make the reward signal less consistent and harder to learn from. This is especially hard with increasing number of objects, as the chances of at least one object being occluded are very high. This is highlighted by the drop in performance from 1 to 2 cubes, compared to the GT reward. Image-based reward calculation for single object manipulation, as in SMORL, is slightly more consistent as occlusions in a single view will not affect the overall reward as much. Adding more viewpoints for the reward calculation might improve these results without increasing inference complexity. We see that in the 3-Cubes environment, our method surpasses SMORL, although SMORL's reward is based on a single object regardless of the number of objects in the environment. This could be as a result of object-object interactions being more significant in this case.

| Method | Success Rate | Success Fraction | Max Obj Dist | Avg Obj Dist | Avg Return |
|---|---|---|---|---|---|
| Ours | $0.765 \pm 0.025$ | $0.875 \pm 0.015$ | $0.037 \pm 0.002$ | $0.026 \pm 0.001$ | -0.210 $\pm$ 0.009 |
| SMORL | $0.838 \pm 0.016$ | $0.911 \pm 0.008$ | $0.038 \pm 0.004$ | $0.025 \pm 0.002$ | -0.320 $\pm$ 0.007 |
| Ours | $0.580 \pm 0.093$ | $0.822 \pm 0.052$ | $0.063 \pm 0.008$ | $0.035 \pm 0.005$ | -0.251 $\pm$ 0.022 |
| SMORL | $0.509 \pm 0.044$ | $0.794 \pm 0.024$ | $0.092 \pm 0.006$ | $0.047 \pm 0.004$ | -0.451 $\pm$ 0.031 |

Table 2: *Performance Metrics: Image-Based Rewards Methods trained and evaluated on the* 2-Cubes *(top) and* 3-Cubes *(bottom) environments. Values calculated on* 400 *random goals per random seed.*

## C ADDITIONAL RESULTS

### C.1 MULTI-OBJECT MANIPULATION

Performance metrics for the 2-Cubes and 3-Cubes are presented in Table 3

| Method | Success Rate | Success Fraction | Max Obj Dist | Avg Obj Dist | Avg Return |
|---|---|---|---|---|---|
| Ours (State) | $0.991 \pm 0.004$ | $0.995 \pm 0.003$ | $0.014 \pm 0.001$ | $0.010 \pm 0.001$ | -0.129 $\pm$ 0.006 |
| SMORL (State) | $0.980 \pm 0.006$ | $0.989 \pm 0.005$ | $0.014 \pm 0.002$ | $0.009 \pm 0.002$ | -0.142 $\pm$ 0.016 |
| Ours (Image) | $0.968 \pm 0.019$ | $0.983 \pm 0.009$ | $0.020 \pm 0.002$ | $0.015 \pm 0.001$ | -0.150 $\pm$ 0.008 |
| Ours (State) | $0.978 \pm 0.006$ | $0.991 \pm 0.002$ | $0.016 \pm 0.001$ | $0.010 \pm 0.001$ | -0.124 $\pm$ 0.007 |
| SMORL (State) | $0.932 \pm 0.022$ | $0.974 \pm 0.009$ | $0.028 \pm 0.005$ | $0.015 \pm 0.002$ | -0.201 $\pm$ 0.011 |
| Ours (Image) | $0.919 \pm 0.008$ | $0.969 \pm 0.004$ | $0.026 \pm 0.002$ | $0.016 \pm 0.001$ | -0.157 $\pm$ 0.007 |

Table 3: *Performance Metrics: GT Reward Methods trained and evaluated on the* 2-Cubes *(top) and* 3-Cubes *(bottom) environments. Values calculated on* 400 *random goals per random seed.*

## C.2 GENERALIZATION

**Number of Objects**: Performance metrics for the compositional generalization to different numbers of objects of an agent trained on the `3-Cubes` environment are presented in Table 4. Additionally, we compare compositional generalization performance with respect to the number of objects seen during training in Table 5. We see that when learning with 3 objects our agent is able to generalize reasonably well to a larger amount of objects. This is not the case with agents trained on 1 or 2 objects, where there is a sharp decay in performance starting from a single additional object. When training on 1 object, the policy lacks the need to perform reasoning between multiple objects in the state, thus it is not surprising it does not generalize to more than a single object. While training on 2 objects does require this type of reasoning, we believe training on 3 objects has such an increase in generalization abilities because the agent encounters more scenarios during training where modeling object interaction and interference is necessary.

| Number of Cubes | Success Rate | Success Fraction | Max Obj Dist | Avg Obj Dist | Avg Return |
|:---:|:---:|:---:|:---:|:---:|:---:|
| 1 | 0.973 | 0.973 | 0.016 | 0.016 | -0.162 |
| 2 | 0.963 | 0.981 | 0.023 | 0.017 | -0.154 |
| **3** | **0.838** | **0.942** | **0.034** | **0.02** | **-0.175** |
| 4 | 0.723 | 0.912 | 0.051 | 0.027 | -0.213 |
| 5 | 0.57 | 0.876 | 0.068 | 0.031 | -0.245 |
| 6 | 0.398 | 0.826 | 0.09 | 0.036 | -0.294 |

Table 4: ***Performance Metrics for Different Number of Cubes than in Training*** – *Our method's performance on different numbers of cubes in the $N$-`cubes` environment, trained on the $3$-`cubes` environment (results in **bold**) with cubes of $6$ different colors. Values are averaged over $400$ episodes with randomly initialized goal and initial configurations.*

| Cubes in Test ↓ / Train → | 3 | 2 | 1 |
|:---:|:---:|:---:|:---:|
| 1 | 0.016 | 0.013 | 0.017 |
| 2 | 0.023 | 0.024 | 0.256 |
| 3 | 0.034 | 0.091 | 0.287 |
| 4 | 0.051 | 0.149 | 0.311 |
| 5 | 0.068 | 0.276 | 0.320 |
| 6 | 0.09 | 0.292 | 0.328 |

Table 5: ***Maximum Object Distance Comparison for Different Number of Cubes than in Training*** – *Our method's performance on different numbers of cubes in the $N$-`cubes` environment. We compare agents trained on the $1, 2, 3$-`cubes` environments with cubes of $6$ different colors. Values are averaged over $400$ episodes with randomly initialized goal and initial configurations.*

**Distracters**: An additional scenario we consider is providing the agent a goal image which contains some cube colors that are not present in the environment. We term these cubes *distracters*. The agent is able to disregard the distracters in the goal image while successfully manipulating the other cubes to their goal locations. A demonstration of these capabilities are available on our website.

**Object Properties**: While we designed our algorithm to facilitate compositional generalization, it is interesting to study its generalization to different object properties. Dealing with novel objects would require generalization from both the DLP and the EIT. We would expect our method to zero-shot generalize to novel objects in case: (1) They are visually similar to objects seen during training. (2) Their physical dynamics are similar to the objects seen during training. To test our hypothesis, we deploy our trained agent in environments including the following modifications: (I) Cuboids obtained by enlarging either the x dimension or both x and y dimensions of the cube. (II) Star shaped objects with the same effective radius of the cubes seen during training. (III) Cubes with different masses than in training. (IV) Cubes in colors not seen in RL training. (V) Cubes in colors not seen in RL training or DLP pre-training. Performance metrics for these cases are presented in Table 6. Visualizations of the modified object environments and how the DLP model perceives them are available in Figure 9.

*Change in Shape* - Based on our study, the DLP model is able to capture the different shaped objects and their location. Observing the reconstruction of the scene, DLP models the objects using the building blocks it knows, which are cubes. Stars are mapped to cubes and cuboids are mapped to a composition of multiple cubes. While this is an interesting form of generalization for image reconstruction, it is not sufficient for inferring changes in dynamics. In cases where the shape does not strongly affect dynamics such as the star and the slightly modified cuboid, the EIT agent still achieves strong performance. When this is not the case there is a significant performance drop, as expected.

*Change in Color* - With a similar study of the DLP reconstruction we witnessed an interesting yet not surprising phenomenon: colors not seen in DLP pretraining are mapped to the closest known color in the latent space. For example brown is mapped to green and orange to yellow, pink to purple. Judging by the non-negligible success rates we hypothesize that EIT generalizes to both control and matching of colors it has not seen in RL training via the DLP latent space. We believe the reason for the drop in performance originates in ambiguity caused by inconsistent mapping of colors. One failure scenario is when the goal object is mapped to a different color than the corresponding state object, due to differences originating in shading or other factors. Another failure scenario is two different objects being mapped to the same color, causing ambiguity in goal specification which the agent is not expected to generalize to. The above could provide an explanation to the fact that our agent performs better with colors not seen in both DLP and RL training than with colors only seen by DLP: the performance more strongly depends on the *combination of colors* than on the ability of DLP to recognize each color individually.

*Change in Mass* - Changes in mass have resulted in the smallest performance drop out of all the scenarios we considered. While the change in mass has a significant affect on dynamics, it does not affect dynamics related to torques (moment of inertia matrix is of the same structure) or agent-object contact. Additionally it does not affect the appearance of objects and therefore does not require generalization from the DLP. We believe our EIT policy is able to generalize well to these changes because it is Markovian and therefore reactive: it does not need to predict the exact displacement of the object in order to infer the direction of its action and can simply react to the unraveling sequence of states.

Our main conclusion from this study is that while zero-shot generalization to objects with different properties is only partial, the non-negligible success rates hint at potential few-shot generalization.

| Configuration | Success Rate | Success Fraction | Max Obj Dist | Avg Obj Dist | Avg Return |
|---|---|---|---|---|---|
| Training | $0.919 \pm 0.008$ | $0.969 \pm 0.004$ | $0.026 \pm 0.002$ | $0.016 \pm 0.001$ | $-0.157 \pm 0.007$ |
| Star | $0.902 \pm 0.002$ | $0.961 \pm 0.002$ | $0.031 \pm 0.005$ | $0.019 \pm 0.001$ | $-0.167 \pm 0.007$ |
| Cuboid $x \cdot = 1.5$ | $0.743 \pm 0.061$ | $0.891 \pm 0.030$ | $0.052 \pm 0.008$ | $0.029 \pm 0.004$ | $-0.228 \pm 0.023$ |
| Cuboid $x \cdot = 2$ | $0.403 \pm 0.079$ | $0.694 \pm 0.056$ | $0.124 \pm 0.020$ | $0.063 \pm 0.010$ | $-0.408 \pm 0.043$ |
| Cuboid $x, y \cdot = 1.5$ | $0.316 \pm 0.077$ | $0.635 \pm 0.060$ | $0.205 \pm 0.037$ | $0.097 \pm 0.017$ | $-0.560 \pm 0.078$ |
| Cuboid $x, y \cdot = 2$ | $0.026 \pm 0.008$ | $0.274 \pm 0.040$ | $0.398 \pm 0.009$ | $0.217 \pm 0.014$ | $-1.089 \pm 0.048$ |
| Colors New to EIT | $0.379 \pm 0.052$ | $0.728 \pm 0.031$ | $0.094 \pm 0.021$ | $0.047 \pm 0.009$ | $-0.306 \pm 0.041$ |
| Colors New to EIT + DLP | $0.550 \pm 0.100$ | $0.810 \pm 0.050$ | $0.089 \pm 0.009$ | $0.042 \pm 0.005$ | $-0.307 \pm 0.026$ |
| Mass $\cdot = 0.05$ | $0.900 \pm 0.004$ | $0.961 \pm 0.001$ | $0.033 \pm 0.002$ | $0.019 \pm 0.001$ | $-0.164 \pm 0.005$ |
| Mass $\cdot = 0.1$ | $0.888 \pm 0.023$ | $0.956 \pm 0.008$ | $0.035 \pm 0.003$ | $0.020 \pm 0.001$ | $-0.169 \pm 0.007$ |
| Mass $\cdot = 10$ | $0.881 \pm 0.014$ | $0.945 \pm 0.008$ | $0.032 \pm 0.002$ | $0.021 \pm 0.001$ | $-0.244 \pm 0.009$ |
| Mass $\cdot = 20$ | $0.751 \pm 0.019$ | $0.876 \pm 0.010$ | $0.061 \pm 0.009$ | $0.035 \pm 0.004$ | $-0.378 \pm 0.016$ |

Table 6: ***Performance Metrics: Zero-shot Generalization to Object Properties*** – *Methods trained from images with GT reward and evaluated on the* `3-Cubes` *environment with red, green and blue cubes. Values calculated on* $400$ *random goals per random seed.*

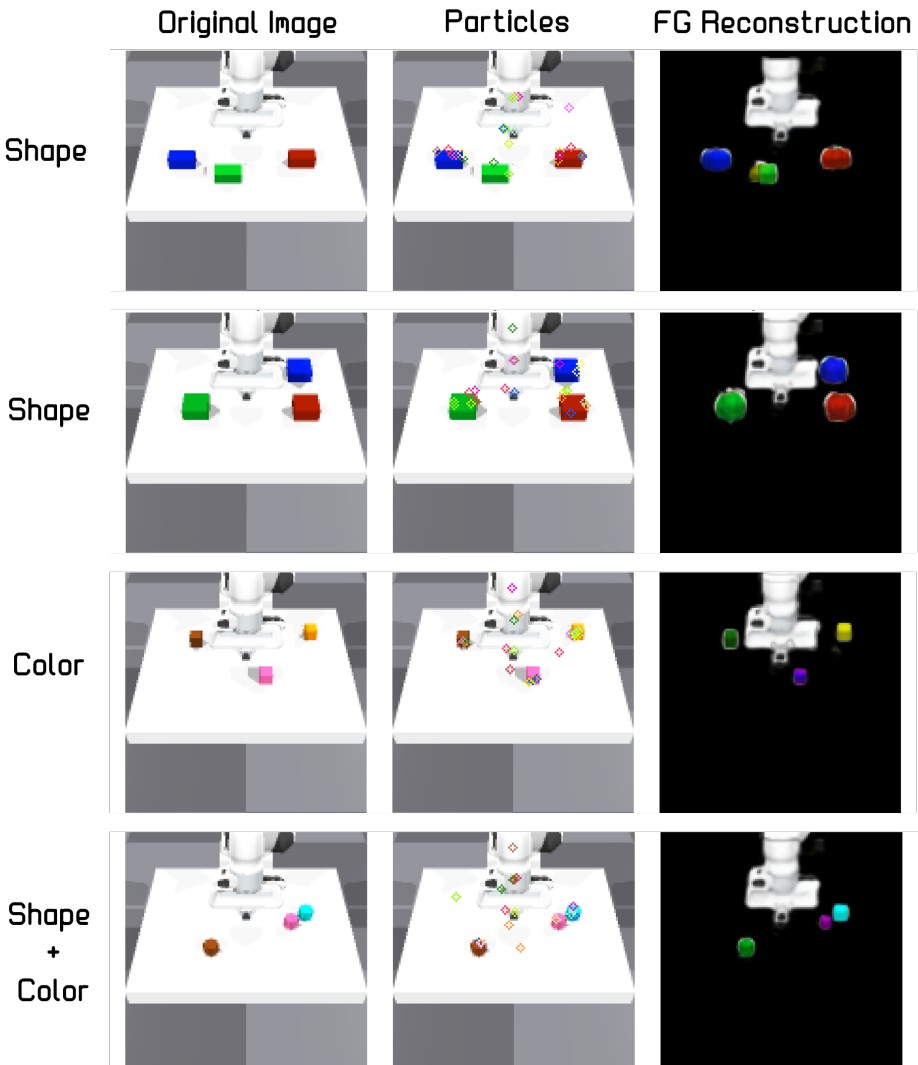

Figure 9: *DLP perception of environments with shape and color modifications of cube objects not seen during training.* **Left to Right**: *Raw Image | Visualization of Particle Locations | Foreground Reconstruction of the DLP Decoder.* **Top to Bottom**: *Cuboid $x \cdot = 2$ | Cuboid $x, y \cdot = 2$ | New Colors | Star Shape & New Colors.*

### C.3 PUSH-T

See Figure 10 for a visualization and performance results on the `Push-T` task.

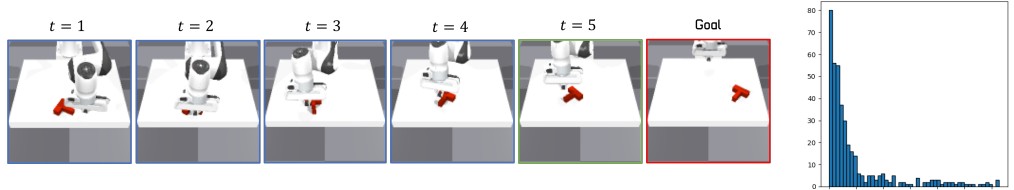

Figure 10: **Left** – *Rollout of an agent trained on the* `Push-T` *task.* **Right** – *Distribution of object angle difference (radians) from goal. Values of* 400 *episodes with randomly initialized goal and initial configurations.*

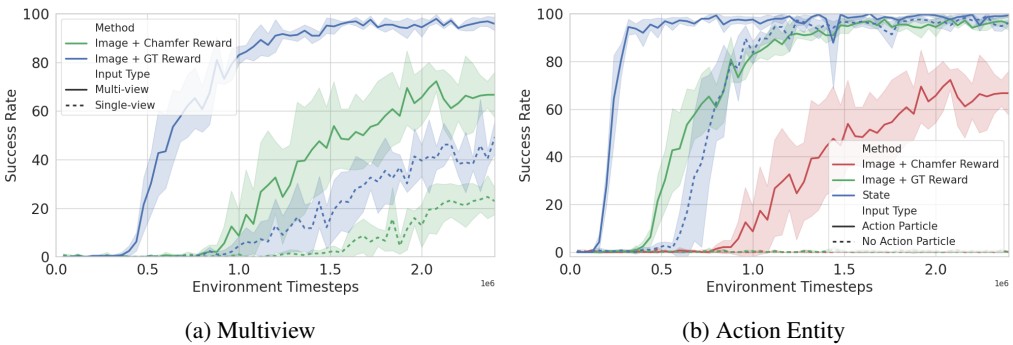

(a) Multiview                 (b) Action Entity

Figure 11: *Success Rate vs. Environment Timesteps - Multiview and Action Entity Ablation* –
*Values calculated based on* 96 *randomly sampled goals.*

## C.4 ABLATION STUDY

We explore how aspects we found to be key to the success of our proposed method effect sample
efficiency. Figure 11a compares our method with multi-view vs. single-view image inputs. We
find that both with GT and image-based reward, multi-view inputs substantially improve sample
efficiency. We believe that the connections formed between particles from different views in the
Transformer blocks makes it easier for the agent to learn the correlations between actions defined in
3D space and latent attributes defined in 2D pixel space. In addition, multiple viewpoints decrease
the degree of partial observability. Figure 11b compares treating the action as a separate input
entity to the Q-function Transformer blocks vs. concatenating the action to the output of the final
Transformer block, before the output MLP. We find that learning the relations between the action and
the state and goal entities via the attention mechanism is crucial to the performance of our method.
Without it, our experiments exhibit decreased sample efficiency in state observations and failure to
learn in the given environment timestep budget with image observations.

Figure 12 presents an ablation of the contribution of DLP's attributes, $z = (z_p, z_s, z_d, z_t, z_f)$, to
the agent's success on the 2-Cubes environment. The position attribute $z_p$ and visual features $z_f$
contain necessary location and entity-identifying information, therefore we do not run experiments
without them. The results show equivalent performance when discarding the depth $z_d$ and trans-
parency $z_t$ attributes as well as without attention masking based on the transparency. The scale
attribute $z_s$, on the other hand, proves to be significant for sample efficiency although it does not
affect final performance.

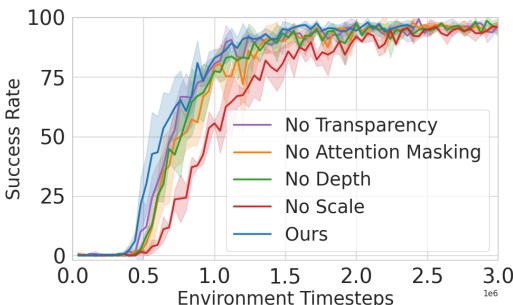

Figure 12: *Success Rate vs. Environment Timesteps - DLP Attribute Ablation* – *Values calculated
based on* 96 *randomly sampled goals.*

Figure 13 compares the performance of our method with DLP vs. Slot-Attention (SA, Locatello
et al. (2020)) as the OCR. On the 1-Cube environment the performance is equivalent (see 13a).
Note that the $(x, y)$ coordinates are not explicit in the SA latent representation. Nevertheless, the

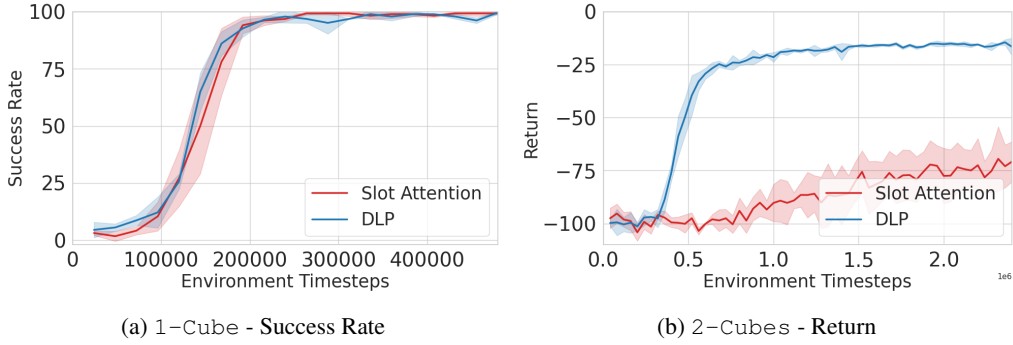

(a) `1-Cube` - Success Rate            (b) `2-Cubes` - Return

Figure 13: ***OCR Performance Ablation*** – *Values calculated based on* 96 *randomly sampled goals.*

EIT is able to infer object location from the slots. These results showcase similar capabilities to the ones presented in the `Push-T` task, where the EIT was able to infer orientation from the DLP latent visual attributes. In the `2-Cubes` environment, despite observing a moderate increase in return during training (see Figure 13b), our method with SA was unable to solve the task, achieving approximately $0\%$ success rates. From an investigation of these experiments and the representations produced by SA, we found that often, both cubes were assigned to a single slot. This led to the agent learning to push both cubes to the middle point between the two cubes' goals. We hypothesize that this behavior is optimal given that the agent can only infer a single location for both objects in the same slot. We were not able to train a SA model which consistently separated the cubes to different slots. Our design choice of utilizing DLP, coupled with training a model with a relatively large number of particles (24) and limiting the capacity of each particle's visual latent features ($z_f \in \mathbb{R}^4$), effectively prevented multiple cubes from being assigned to a single particle.

## D    IMPLEMENTATION DETAILS AND HYPER-PARAMETERS

In this section, we provide extensive implementation details in addition to the open-source code that can be found in the official repository: `https://github.com/DanHrmti/ECRL`.

### D.1    ENVIRONMENT

We implement our environments with IsaacsGym (Makoviychuk et al., 2021), by adapting code from IsaacGymEnvs[3] and OSCAR[4].

**Ground-truth State** Denote $s_i = (x_i, y_i)$, $g_i = (x_i^g, y_i^g)$ the xy coordinates of the state and goal of entity $i$ respectively. The input to the networks in the structured methods are two sets of vectors $\{v_i\}_{i=1}^N$, $v_i = [s_i, \text{one-hot}\,(i|N)] \in \mathbb{R}^{2+N}$, $\{u_i\}_{i=1}^N$, $u_i = [g_i, \text{one-hot}\,(i|N)] \in \mathbb{R}^{2+N}$, $[\cdot]$ denoting concatenation, where the one-hot vectors serve as entity-identifying features. In the unstructured case, the input is $[s_1, s_2, ..., s_N, g_1, g_2, ..., g_N]$.

**Ground-truth Reward** The reward calculated from the ground-truth state of the system, which we refer to as the ground-truth (GT) reward, is the mean negative $L_2$ distance between each cube and its desired goal position on the table:

$$r_t = -\frac{1}{N} \sum_{i=1}^N \frac{1}{L} \left\| g_i^d - g_i^a \right\|_2,  \tag{2}$$

where $g_i^d$ and $g_i^a$ denote the desired and achieved goal for object $i$ respectively, $N$ the number of objects, $r_t$ the immediate reward at timestep $t$ and $L$ a normalization constant for the reward corresponding to the dimensions of the table.

---

[3]`https://github.com/NVIDIA-Omniverse/IsaacGymEnvs`
[4]`https://github.com/NVlabs/oscar`

**Image-Based Reward** The reward calculated from the DLP OCR for our method is the negative GDAC distance (see Eq. 1) between state and goal sets of particles, averaged over viewpoints:

$$r_t = -\frac{1}{K} \sum_{k=1}^{K} Dist_{GDAC} \left( \{p_m^k\}_{m=1}^{M}, \{q_m^k\}_{m=1}^{M} \right), \tag{3}$$

where we use $D_1(x, y) = \left\| z_p^x - z_p^y \right\|_1$ and $D_2(x, y) = \left\| z_f^x - z_f^y \right\|_2$ in the GDAC distance, $z_p^{(\cdot)}$, $z_f^{(\cdot)}$ denoting DLP latent attribute $z_p, z_f$ of particle $(\cdot)$ respectively. We filter out particles that do not correspond to cubes (see section B.1) for the distance calculation. When a particle has no match (i.e. $\min_y \left\| z_f^x - z_f^y \right\|_2 > C$), a negative bonus is added to the reward to avoid "reward hacking" by pushing blocks off the table or occluding them intentionally.

**Evaluation Metrics** We evaluate the performance of the different methods based on the following:

*Success*: $\mathbb{I}\left(\sum_{i=1}^{N} \mathbb{I}\left(\left\| g_i^d - g_i^a \right\|_2 < R\right) = N\right)$, all $N$ objects are at a threshold distance from their desired goal. $R$ denotes the success threshold distance and is slightly smaller than the effective radius of a cube. $\mathbb{I}$ denotes the indicator function. This metric most closely captures task success, but does not capture intermediate success or timestep efficiency.

*Success Fraction*: $\frac{1}{N} \sum_{i=1}^{N} \mathbb{I}\left(\left\| g_i^d - g_i^a \right\|_2 < R\right)$, fraction of objects that reach individual success.

*Maximum Object Distance*: $\max_i \left\{ \left\| g_i^d - g_i^a \right\|_2 \right\}$, largest distance of an object from its desired goal.

*Average Object Distance*: $\frac{1}{N} \sum_{i=1}^{N} \left\| g_i^d - g_i^a \right\|_2$, average distance of objects from their desired goal.

*Average Return*: $\frac{1}{T} \sum_{t=1}^{T} r_t$, the immediate GT reward averaged across timesteps, where $T$ is the evaluation episode length. A high average return means that the agent solved the task quickly.

## D.2 REINFORCEMENT LEARNING

We implement our RL algorithm with code adapted from `stable-baselines3` (Raffin et al., 2021). Specifically, we use TD3 (Fujimoto et al., 2018) with HER (Andrychowicz et al., 2017). We use $\varepsilon$-greedy and Gaussian action noise for exploration, that decays to half its initial value with training progress, similar to Zhou et al. (2022). We use Adam for neural network optimization. Related hyper-parameters can be found in Table 7 and Table 8.

| Learning Rate | 5e-4 |
|---|---|
| Batch Size | 512 |
| $\gamma$ | 0.98 |
| $\tau$ | 0.05 |
| # Episodes Collected per Training Loop | 16 |
| Update-to-Data Ratio | 0.5 |
| HER Ratio | 0.8 |
| Exploration Action Noise $\sigma$ | 0.2 |
| Exploration $\varepsilon$ | 0.3 |

Table 7: General hyper-parameters used for RL training.

| Number of Cubes | 1 | 2 | 3 |
|---|---|---|---|
| Episode Horizon | 30 | 50 | 100 |
| Replay Buffer Size | 100000 | 100000 | 200000 |

Table 8: Environment specific hyper-parameters used for RL training.

For the Entity Interaction Transformer (EIT) we adapted components from DDLP's Particle Interaction Transformer (PINT, Daniel & Tamar (2023)), which is based on a Transformer decoder architecture and utilizing the open-source minGPT (Karpathy, 2021) code base. Related hyper-parameters can be found in Table 9.

| Attention Dimension | 64 |
|---|---|
| Attention Heads | 8 |
| MLP Hidden Dimension | 256 |
| MLP Number of Layers | 3 |

Table 9: Hyper-parameters for the EIT architecture.

**Attention Masking**: The DLP model extracts a fixed number of particles, which often include particles that do not represent objects in the image. These particles are assigned low transparency ($z_t$) values by the DLP model as to not affect the reconstruction quality. We disregard these particles in our policy and Q-function by directly masking the attention entries related to them. We found that this slightly improves sample efficiency but is not crucial to performance as the EIT is able to learn to disregard these particles by assigning them very low attention values.

Policies for the unstructured baselines have 5 layer MLPs with hidden dimension 256.

### D.3 PRE-TRAINED IMAGE REPRESENTATIONS

In this section, we detail the various *unsupervised* pre-trained image representation methods used in this work. We begin with the non-object-centric baselines, i.e., methods that given an image $I \in \mathbb{R}^{H \times W \times C}$, encode a single-vector representation $z \in \mathbb{R}^D$, where $D$ is the latent dimension, of the entire input image. Then, we describe the object-centric representation (OCR) method that provides a structured latent representation $z \in \mathbb{R}^{K \times d}$ of a given image $I$, where $K$ is the number of entities in the scene, each described by latent features of dimension $d$.

**Data**: We collect $600,000$ images from 2 viewpoints by interacting with the environment using a random policy for $300,000$ timesteps. For all methods, we use RGB images at a resolution of $128 \times 128$, i.e., $I \in \mathbb{R}^{128 \times 128 \times 3}$.

**Variational Autoencoder (VAE)**: We train a $\beta$-VAE (Higgins et al., 2017) with a latent bottleneck of size $D = 256$, i.e., each image $I$ is encoded as $z \in \mathbb{R}^{256}$. We adopt a similar autoencoder architecture as Rombach et al. (2022) based on the open-source implementation[5] and add a 2-layer MLP with 512 hidden units after the encoder and before the decoder to ensure the latent representation is of dimension 256. We use $\beta = 1e - 10$, a batch size of 16 and an initial learning rate of $2e - 4$ which is gradually decayed with a linear schedule. The model is trained for 40 epochs with a perceptual reconstruction loss and $L_1$ pixel-wise loss, similarly to Rombach et al. (2022), and we keep the default values for the rest of the hyper-parameters.

**Deep Latent Particles (DLP)**: We train a DLPv2 (Daniel & Tamar, 2023) using the publicly available code base[6] as our unsupervised OCR model. **We modify the DLP model to have background particle features of dimension $1$, and discard the background particle for RL purposes.** The background is static in our experiments and setting its latent particle to have a single feature is meant to limit its capacity to capture changing parts of the scene such as the objects or the agent. Recall that DLP provides a disentangled latent space structured as a set of foreground particles $z = \left\{(z_p, z_s, z_d, z_t, z_f)_i\right\}_{i=0}^{K-1} \in \mathbb{R}^{K \times (6+l)}$, where $K$ is the number of particles. Figure 14 illustrates an example of the object-centric decomposition for a single image using a DLP model pre-trained on our data. We keep the default recommended hyper-parameters and report the data-specific hyper-parameters in Table 10. Note that our DLP model represents an image $I$ by a total of $K \times (6 + l) = 24 * (6 + 4) = 240$ latent features.

**Slot-Attention**: For the OCR ablation study we train a Slot-Attention (Locatello et al., 2020) model with 10 slots, each of size $D = 64$, i.e., each image $I$ is encoded as $z \in \mathbb{R}^{10 \times 64}$. We use the implementation from `https://github.com/HHousen/object-discovery-pytorch`, and keep most of the hyper-parameters similar, corresponding to the ones used in the original paper, and we provide the set of hyper-parameters in our code repository. We performed multiple training runs with each set of hyper-parameters and took the run that yielded the best object separation to slots, similar to the training procedure in Wu et al. (2022).

---

[5]`https://github.com/CompVis/latent-diffusion`
[6]`https://github.com/taldatech/ddlp`

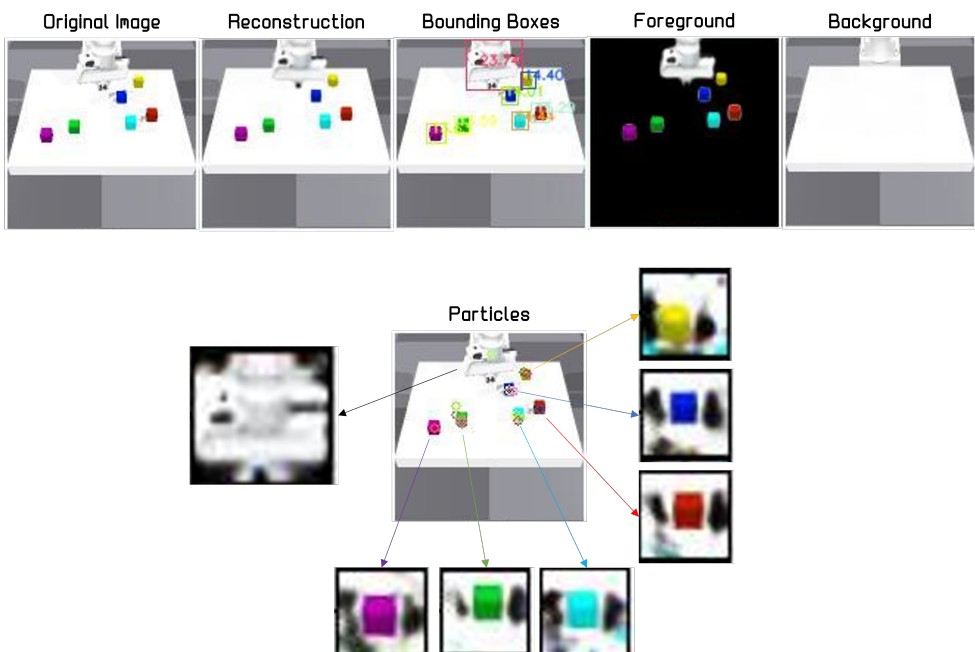

Figure 14: *Object-centric Decomposition with DLP* – *DLP decomposes a single image into latent particles, each characterized by attributes including position (keypoints in the images), scale (bounding boxes), and visual appearance features around the keypoint (displayed as decoded glimpses from these features).*

| | |
|---|---|
| Batch Size | 64 |
| Posterior KP $K$ | 24 |
| Prior KP Proposals $L$ | 32 |
| Reconstruction Loss | MSE |
| $\beta_{KL}$ | 0.1 |
| Prior Patch Size | 16 |
| Glimpse Size $S$ | 32 |
| Feature Dim $m$ | 4 |
| Background Feature Dim $m_{\text{bg}}$ | 1 |
| Epochs | 60 |

Table 10: Hyper-parameters used for the Deep Latent Particles (DLP) object-centric model.

### D.4 SMORL REIMPLEMENTATION

We re-implement SMORL (Zadaianchuk et al., 2020) based on the official implementation[7], using the same code-base as we used for the EIT for the SMORL attention architecture. SMORL specific hyper-parameters are detailed in Table 11. We extend SMORL to multiple views, which includes modifications to several aspects of the algorithm:

**Attention Architecture**: We extend SMORL's attention policy by adding a goal-conditioned and goal-unconditional attention block for the additional view. The outputs of attention layers from both views are concatenated and fed to an MLP, as in the single-view version. An outline of SMORL's single-view attention-based architecture is described in Figure 15.

---

[7]https://github.com/martius-lab/SMORL

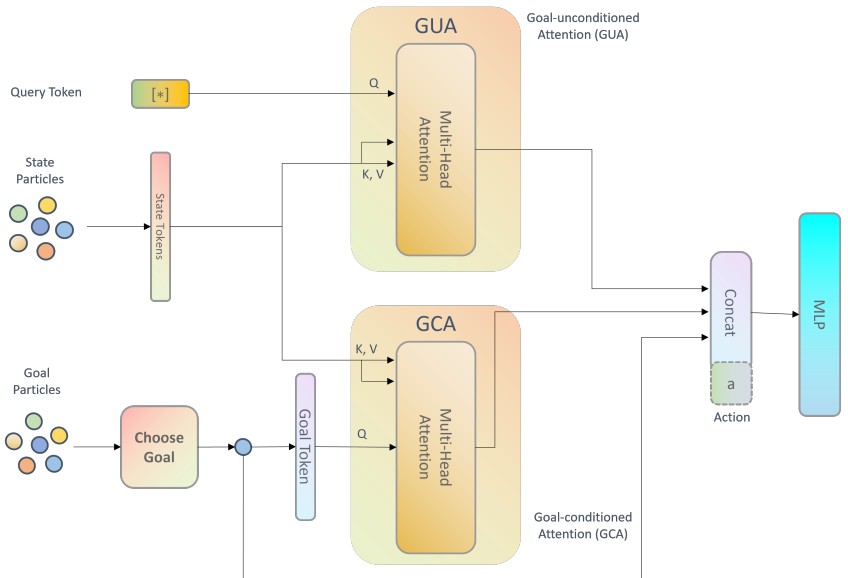

Figure 15: *Outline of SMORL's Attention Architecture - The policy is conditioned on the goal by choosing a single goal particle and feeding it to a cross-attention block between the goal particle and the state particles. In parallel, cross-attention between a learned particle and the state particles is performed to extract features from the state that are not goal-dependant. The outputs of the two attention layers are then concatenated to the original goal particle and fed to an MLP to produce the action. For the Q-function, the input action is additionally concatenated to the output of the attention to produce the value.*

**Selecting a Single Goal**: SMORL decomposes the multi-object goal-conditioned task to single objects by selecting a single goal at a time, and rewarding the agent with respect to this sub-goal alone. Working with multiple views requires selecting a goal particle corresponding to the same object from both viewpoints, which requires explicit matching. We do this by selecting a goal particle from one viewpoint and choosing the closest matching particle from the second viewpoint based on the $L_2$ distance in latent attribute $z_f$.

**Reward**: The image-based reward calculated from the DLP OCR for SMORL is the negative $L_2$ distance in attribute $z_p$ between the goal particle to the closest matching state particle based on attribute $z_f$, averaged over viewpoints:

$$r_t = -\frac{1}{K}\sum_{k=1}^{K}\left\|z_p^{g^k} - z_p^{s_{m_k}^k}\right\|_2, \quad m_k = \arg\min_m \left\|z_f^{g^k} - z_f^{s_m^k}\right\|_2, \tag{4}$$

$g^k$ denoting the goal particle from view k and $s_m^k$ denoting particle $m$ from view $k$. When there is no match for the goal particle in viewpoint $k$ (i.e. $\min_m \left\|z_f^{g^k} - z_f^{s_m^k}\right\|_2 > C$), the minimal reward is given for that view. Note that this reward is a special case of the Chamfer reward we define in this work, with the goal set consisting of a single entity per view.

| | |
|---|---|
| Attention Dimension | 64 |
| Unconditional Attention Heads | 8 |
| Goal-conditioned Attention Heads | 8 |
| MLP Hidden Dimension | 256 |
| MLP Layers | 4 |
| Scripted Meta-policy Steps | 15 |

Table 11: SMORL hyper-parameters.

# E ATTENTION IN RL POLICIES - COMPARISON TO PREVIOUS WORK

In this section, we compare our use of attention to two previous approaches, Zhou et al. (2022) which is state-based and SMORL (Zadaianchuk et al., 2020) which is image-based.

Zhou et al. (2022) also propose a Transformer-based policy. They define an entity as the concatenation of each object's state, goal, and the state of the agent (and the action when we consider the input to the Q-function). This requires explicitly matching between entities in state and goal as well as identifying the agent, which is trivial when working with GT state observations. When learning from OCRs of images, this is not at all trivial. Matching is not always possible due to lack of a one-to-one match or occlusion, which also limits the use of multiple viewpoints. We tackle this in our EIT by using a cross-attention block for goal-conditioning. Additionally, we learn which particles correspond to the agent *implicitly* through the RL objective. Figure 16 describes the differences between their definition of an input entity to ours.

SMORL uses an OCR of images and does not require matching between entities in the single-view case. The goal-conditioned attention in their architecture (see Figure 15) matches a single goal particle to the relevant state particle via cross-attention. Different from us, the attention mechanism is used only to extract sub-goal specific entities from the set of state entities, and does not explicitly model relationships between the different entities in the state. We do this explicitly by incorporating self-attention Transformer blocks in our architecture.

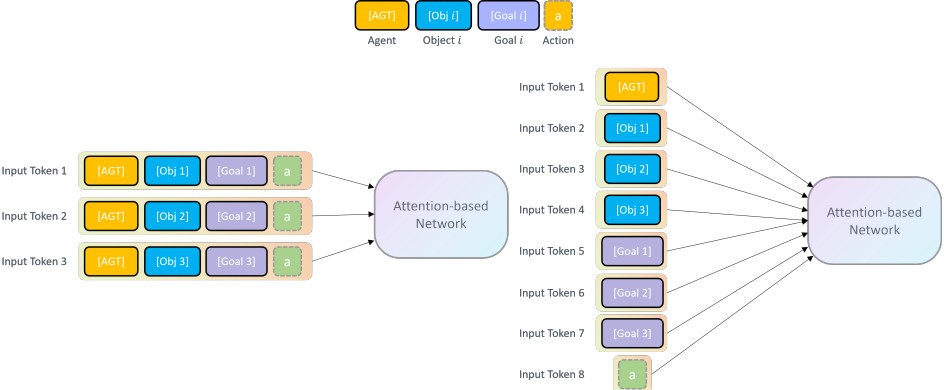

Figure 16: **Entity Definition Comparison** *Left – description of an input token defined by  Zhou et al. (2022), where each token is a concatenation of the object and corresponding goal as well as global entities such as the agent and action; Right – Our definition of input tokens, where each entity is treated as a separate token.*

# F COMPOSITIONAL GENERALIZATION THEORY

We begin by defining a notion of compositionally generalizing functions F.1.1 and provide an example of such a class of functions F.1.2. We additionally provide an example of a case where a class of functions can accurately approximate another class of functions up to $N$ entities but does not generalize to increasing number of entities F.1.3. Following these, we prove our main theorem F.2. We then present modifications to the assumptions of our main theorem to obtain a result F.3 that adheres to Definition 3.

Relating the theorems to compositionally generalizing policies, we show that an $\varepsilon$-optimal Q-function for $M$ objects is also approximately optimal for $M + k$ objects under our structural assumptions. This implies that a policy that is trained to maximize this Q-function with $M$ objects should also achieve high returns for $M + k$ objects, i.e. achieve zero-shot compositional generalization.

## F.1 COMPOSITIONALLY GENERALIZING FUNCTIONS

### F.1.1 DEFINITION

In the following, we define a notion of *compositionally generalizing* functions:

**Definition 3.** Denote the space of variable-sized sets of entities $\mathcal{S} = \cup_{N \in \mathbb{N}} \mathcal{S}^N$ where $\mathcal{S}^N = \{\{s_i\}_{i=1}^N | s_i \in \tilde{\mathcal{S}}\}$, $\tilde{\mathcal{S}}$ being the space of a single entity. Denote $\mathcal{S}^{<N} \subset \mathcal{S}$ the subspace containing sets of up to size $N$.
A class of functions $\mathcal{C} : \mathcal{S} \to \mathbb{R}$ is said to admit *compositional generalization* on $\mathcal{X} \subseteq \mathcal{S}$ if there exists $N$ such that for any $f^*, f \in \mathcal{C}$ that satisfy $\|f^*(x) - f(x)\| < \varepsilon, \quad \forall x \in \mathcal{X}^{<N}$, we have that for any $M > 0$ and $x \in \mathcal{X}^{<N+M}$: $\|f^*(x) - f(x)\| < (C_1 + C_2 \cdot M) \cdot \varepsilon, \quad \forall x \in \mathcal{X}$ where $C_1, C_2$ are constants that do not depend on $M, \varepsilon$.

Put simply, our definition of compositional generalization asks that adding additional $M$ objects to the problem incurs an error that is at most linear in $M$.

### F.1.2 EXAMPLE

An example of a class of compositionally generalizing functions is the class of *DeepSets*-style (Zaheer et al., 2017) function approximators, which are an aggregation of functions defined on single entities:

**Theorem 4.** *Let $\mathcal{C}_{\mathcal{DS}} : \mathcal{S} \to \mathbb{R}$ be the class of functions of the form: $Q(s) = \frac{1}{N} \sum_{i=1}^N v(s_i)$, where $v : \tilde{\mathcal{S}} \to \mathbb{R}$ is a function that operates on single entities. Then $\mathcal{C}_{\mathcal{DS}}$ admits compositional generalization on $\mathcal{S}$.*

*Proof.* Assume $Q^*, \hat{Q} \in \mathcal{C}_{\mathcal{DS}}$ satisfy $\|Q^*(s) - \hat{Q}(s)\| < \varepsilon, \quad \forall s \in \mathcal{S}^{<N}$.
From the case of $N = 1$ we have:

$$\|Q^*(s) - \hat{Q}(s)\| < \|v^*(s) - \hat{v}(s)\| < \varepsilon, \quad \forall s \in \tilde{\mathcal{S}}.$$

From the above equation we obtain $\forall N \in \mathbb{N}$:

$$\|Q^*(s) - \hat{Q}(s)\| = \|\frac{1}{N}\sum_{i=1}^N v^*(s_i) - \frac{1}{N}\sum_{i=1}^N \hat{v}(s_i)\| \le \frac{1}{N}\sum_{i=1}^N \|v^*(s_i) - \hat{v}(s_i)\| < \frac{1}{N}\sum_{i=1}^N \varepsilon = \varepsilon, \quad \forall s \in \mathcal{S}^N.$$

Thus we have that by Definition 3, $\mathcal{C}_{\mathcal{DS}}$ admits compositional generalization on $\mathcal{S}$ with $C_1 = 1, C_2 = 0$. $\square$

### F.1.3 NON-GENERALIZING $\hat{Q}$ EXAMPLE

Assuming $Q^*$ has a self-attention structure, Theorem 2 shows that if we obtained an $\varepsilon$-optimal $\hat{Q}$ for $1, \ldots, M$ objects where $\hat{Q}$ also has a self-attention structure, then the sub-optimality w.r.t. $M + k$ objects grows at most linearly in $M + k$. This raises the question: are there $\hat{Q}$ structures that lack the

compositional generalization quality in this case? In this section we provide a simple example for a structure that can accurately approximate a self-attention $Q^*$ for 2 objects but does not generalize to increasing number of objects.

Consider the following $\hat{Q}$ structure:

$\forall S \in \mathcal{S}^N, \forall a \in \mathcal{A}, \forall N \in \mathbb{N}$: $\hat{Q}(s_1, ..., s_N, a) = \frac{1}{N} \sum_{i=1}^{N} \tilde{Q}_i(s_1, ..., s_N, a)$, where $\tilde{Q}_i(s_1, ..., s_N, a) = \frac{1}{\sum_{j=1}^{N} \alpha(s_i, s_j, a)} \sum_{j=1}^{\tilde{N}} \alpha(s_i, s_j, a) v(s_j, a)$, $\alpha(\cdot) \in \mathbb{R}^+$.

We denote $\tilde{N} = \min\{2, N\}$ where the indices $j = 1, \ldots, N$ are ordered by the value of $v(s_j, a)$ in increasing order. Note that this operation is still invariant to permutations of the input state $s = \{s_i\}_{i=1}^{N}$.

For $N = 1, 2$, this structure is identical to the one assumed in Theorem 2 which is that of $Q^*$ and therefore there exists a $\hat{Q}$ such that: $|\hat{Q} - Q^*| = 0$. In this case, both $v = v^*$ and $\alpha = \alpha^*$.

For $N > 2$ on the other hand, the approximation error cannot be bounded by $\varepsilon$ as $\hat{Q}$ does not account for the entire set of states:

$$\left| \hat{Q}(s_1, ..., s_N, a) - Q^*(s_1, ..., s_N, a) \right| = \left| \frac{1}{N} \sum_{i=1}^{N} \tilde{Q}_i(s_1, ..., s_N, a) - \tilde{Q}_i^*(s_1, ..., s_N, a) \right| =$$

$$= \left| \frac{1}{N} \sum_{i=1}^{N} \frac{1}{\sum_{j=1}^{N} \alpha(s_i, s_j, a)} \sum_{j=1}^{2} \alpha(s_i, s_j, a) v(s_j, a) - \frac{1}{\sum_{j=1}^{N} \alpha^*(s_i, s_j, a)} \sum_{j=1}^{N} \alpha^*(s_i, s_j, a) v^*(s_j, a) \right| =$$

$$= \left| \frac{1}{N} \sum_{i=1}^{N} \sum_{j=1}^{2} \left[ \frac{\alpha(s_i, s_j, a)}{\sum_{j=1}^{N} \alpha(s_i, s_j, a)} v(s_j, a) - \frac{\alpha^*(s_i, s_j, a)}{\sum_{j=1}^{N} \alpha^*(s_i, s_j, a)} v^*(s_j, a) \right] - \sum_{j=3}^{N} \frac{\alpha^*(s_i, s_j, a)}{\sum_{j=1}^{N} \alpha^*(s_i, s_j, a)} v^*(s_j, a) \right| =$$

$$= \left| \frac{1}{N} \sum_{i=1}^{N} \sum_{j=3}^{N} \frac{\alpha^*(s_i, s_j, a)}{\sum_{j=1}^{N} \alpha^*(s_i, s_j, a)} v^*(s_j, a) \right| \geq 0$$

The approximation error is not zero for all inputs $(s_1, ..., s_N, a)$ so long as $v^* > 0$ for some state $s_i \in \tilde{\mathcal{S}}$ and action $a$, which is true unless $Q^*(s_1, ..., s_N, a) = 0$, $\forall s \in \mathcal{S}^N, \forall a \in \mathcal{A}$.

This example illustrates how some function approximators are not well-suited for compositional generalization. Although the approximated model perfectly fits the training distribution, containing up to 2 objects in this case, it does not fit test distributions with more objects.

Relating this to Definition 3, if we consider a class of functions that contains both $\hat{Q}$ and $Q^*$ described above, the example illustrates that this class *does not* admit compositional generalization.

## F.2 MAIN THEOREM PROOF

### F.2.1 LEMMAS

We start by showing that if two positive functions $0 < f(s), g(s)$ normalized by a sum of their values over a set of inputs $\{s_j\}_{j=1}^{N}$ of size $M$ are $\delta$-close to each other, the difference in their normalized value under a set of size $2M - 1$ is bounded by $2\delta$.

**Lemma 5.** *If* $\left| \frac{f(s_i)}{\sum_{j=1}^{N} f(s_j)} - \frac{g(s_i)}{\sum_{j=1}^{N} g(s_j)} \right| < \delta$, $0 < f(s), g(s)$, $\forall s_i, s_j$, $\forall N \in [1, M]$ *then for* $k \in [1, M-1]$, $i \in [1, M]$:

$$\left| \frac{f(s_i)}{\sum_{j=1}^{M+k} f(s_j)} - \frac{g(s_i)}{\sum_{j=1}^{M+k} g(s_j)} \right| \leq 2\delta$$

*Proof.*

$$(*) \quad \left| \frac{f(s_i)}{\sum_{j=1}^{N} f(s_j)} - \frac{g(s_i)}{\sum_{j=1}^{N} g(s_j)} \right| = \left| \frac{f(s_i) \sum_{j=1}^{N} g(s_j) - g(s_i) \sum_{j=1}^{N} f(s_j)}{\left( \sum_{j=1}^{N} f(s_j) \right) \left( \sum_{j=1}^{N} g(s_j) \right)} \right| =$$

$$= \left| \frac{\sum_{j \neq i, j=1}^{N} [f(s_i) g(s_j) - g(s_i) f(s_j)]}{\left( \sum_{j=1}^{N} f(s_j) \right) \left( \sum_{j=1}^{N} g(s_j) \right)} \right| < \delta$$

For $k \in [1, M-1]$, $i \in [1, M]$:

$$\left| \frac{f(s_i)}{\sum_{j=1}^{M+k} f(s_j)} - \frac{g(s_i)}{\sum_{j=1}^{M+k} g(s_j)} \right| = \left| \frac{\sum_{j \neq i, j=1}^{M+k} [f(s_i) g(s_j) - g(s_i) f(s_j)]}{\left( \sum_{j=1}^{M+k} f(s_j) \right) \left( \sum_{j=1}^{M+k} g(s_j) \right)} \right| \leq$$

$$\leq \left| \frac{\sum_{j \neq i, j=1}^{M} [f(s_i) g(s_j) - g(s_i) f(s_j)]}{\left( \sum_{j=1}^{M+k} f(s_j) \right) \left( \sum_{j=1}^{M+k} g(s_j) \right)} \right| + \left| \frac{\sum_{j=M+1}^{M+k} [f(s_i) g(s_j) - g(s_i) f(s_j)]}{\left( \sum_{j=1}^{M+k} f(s_j) \right) \left( \sum_{j=1}^{M+k} g(s_j) \right)} \right| \underset{0 < f(s), g(s)}{\leq}$$

$$\leq \left| \frac{\sum_{j \neq i, j=1}^{M} [f(s_i) g(s_j) - g(s_i) f(s_j)]}{\left( \sum_{j=1}^{M} f(s_j) \right) \left( \sum_{j=1}^{M} g(s_j) \right)} \right| + \left| \frac{\sum_{j=M+1}^{M+k} [f(s_i) g(s_j) - g(s_i) f(s_j)]}{\left( f(s_i) + \sum_{j=M+1}^{M+k} f(s_j) \right) \left( g(s_i) + \sum_{j=M+1}^{M+k} g(s_j) \right)} \right| \underset{(*)}{\leq} 2\delta \Rightarrow$$

$$\Rightarrow \left| \frac{f(s_i)}{\sum_{j=1}^{M+k} f(s_j)} - \frac{g(s_i)}{\sum_{j=1}^{M+k} g(s_j)} \right| \leq 2\delta$$

$\square$

In the following two lemmas we bound the difference between two weighted sums of single-input functions applied individually on a set of inputs $\{s_i\}_{i=1}^{N}$ assuming the weights are $\delta$-close and the function values are $\varepsilon$-close.

**Lemma 6.** *If $\alpha, \beta, v, u \geq 0$ and $|\alpha - \beta| < \delta$, $|v - u| < \varepsilon$, then:*

$$\boxed{|\alpha v - \beta u| < \frac{\alpha + \beta}{2} \varepsilon + \frac{v + u}{2} \delta}$$

*Proof.*

$$|\alpha v - \beta u| = |\alpha v - \alpha u - \beta \cdot u + \alpha u| < |\alpha v - \alpha u| + |\alpha u - \beta \cdot u| = \alpha |v - u| + u |\alpha - \beta| < \alpha \varepsilon + u \delta$$

$$|\alpha v - \beta u| = |\alpha v - \beta \cdot v - \beta \cdot u + \beta \cdot v| < |\beta v - \beta u| + |\alpha v - \beta \cdot v| = \beta |v - u| + v |\alpha - \beta| < \beta \varepsilon + v \delta$$

Combining the above two inequalities we get:

$$|\alpha v - \beta u| < \frac{\alpha + \beta}{2} \varepsilon + \frac{v + u}{2} \delta$$

$\square$

**Lemma 7.** *Let $F(s_1, ..., s_N) = \sum_{i=1}^{N} \frac{f(s_i) g(s_i)}{\sum_{j=1}^{N} f(s_j)}$ and $F^*(s_1, ..., s_N) = \sum_{i=1}^{N} \frac{f^*(s_i) g^*(s_i)}{\sum_{j=1}^{N} f^*(s_j)}$, and*

*assume $\left| \frac{f(s_i)}{\sum_{j=1}^{N} f(s_j)} - \frac{f^*(s_i)}{\sum_{j=1}^{N} f^*(s_j)} \right| < \delta$, $|g(s_i) - g^*(s_i)| < \varepsilon$, $0 \leq g(s_i), g^*(s_i) \leq C$,*

*$0 < f(s), g(s), f^*(s), g^*(s)$. Then:*

$$\boxed{|F(s_1, ..., s_N) - F^*(s_1, ..., s_N)| \leq \varepsilon + N \cdot C \cdot \delta}$$

*Proof.*

$$|F\left(s_1,...,s_N\right) - F^*\left(s_1,...,s_N\right)| = \left| \sum_{i=1}^{N} \frac{f\left(s_i\right)g\left(s_i\right)}{\sum_{j=1}^{N} f\left(s_j\right)} - \frac{f^*\left(s_i\right)g^*\left(s_i\right)}{\sum_{j=1}^{N} f^*\left(s_j\right)} \right| \le$$

$$\le \sum_{i=1}^{N} \left| \frac{f\left(s_i\right)}{\sum_{j=1}^{N} f\left(s_j\right)} g\left(s_i\right) - \frac{f^*\left(s_i\right)}{\sum_{j=1}^{N} f^*\left(s_j\right)} g^*\left(s_i\right) \right| = (*)$$

Using Lemma 6 we get:

$$(*) < \sum_{i=1}^{N} \frac{\frac{f(s_i)}{\sum_{j=1}^{N} f(s_j)} + \frac{f^*(s_i)}{\sum_{j=1}^{N} f^*(s_j)}}{2} \varepsilon + \frac{g\left(s_i\right) + g^*\left(s_i\right)}{2} \delta =$$

$$= \frac{\frac{\sum_{i=1}^{N} f(s_i)}{\sum_{j=1}^{N} f(s_j)} + \frac{\sum_{i=1}^{N} f^*(s_i)}{\sum_{j=1}^{N} f^*(s_j)}}{2} \varepsilon + \sum_{i=1}^{N} \frac{g\left(s_i\right) + g^*\left(s_i\right)}{2} \delta \le$$

$$\le \varepsilon + \sum_{i=1}^{N} C \cdot \delta = \varepsilon + N \cdot C \cdot \delta$$

□

### F.2.2 PROOF

We now prove Theorem 2.

*Proof.* Using $0 \le r \le 1$ and the discounted reward definition we get that $\forall N$:

$$0 \le \hat{Q}\left(s_1,...,s_N,a\right) \le \frac{1}{1-\gamma}$$

By definition of the structure of the Q-function and setting $N = 1$ we get that $\forall s \in \tilde{\mathcal{S}}$:

$$\hat{Q}\left(s,a\right) = \tilde{Q}\left(s,a|s\right) = v\left(s,a\right) \Rightarrow 0 \le v\left(s,a\right) \le \frac{1}{1-\gamma}$$

Again using the definition of the structure of the Q-function we get that $\forall N$:

$$\tilde{Q}_i\left(s_1,...,s_N,a\right) = \frac{1}{\sum_{j=1}^{N} \alpha\left(s_i,s_j,a\right)} \sum_{j=1}^{N} \alpha\left(s_i,s_j,a\right) v\left(s_j,a\right) \le$$

$$\le \frac{1}{\sum_{j=1}^{N} \alpha\left(s_i,s_j,a\right)} \sum_{j=1}^{N} \alpha\left(s_i,s_j,a\right) \frac{1}{1-\gamma} = \frac{1}{1-\gamma}$$

Repeating the above for the optimal Q-function we obtain $\forall N$:

$$0 \le v\left(s,a\right) \le \frac{1}{1-\gamma}, \ 0 \le v^*\left(s,a\right) \le \frac{1}{1-\gamma} \tag{5}$$

$$0 \le \tilde{Q}_i\left(s_1,...,s_N,a\right) \le \frac{1}{1-\gamma}, \ 0 \le \tilde{Q}_i^*\left(s_1,...,s_N,a\right) \le \frac{1}{1-\gamma} \tag{6}$$

Setting $N = 1$ and using the $\varepsilon$-optimality assumption:

$$\left| \tilde{Q}_i\left(s,a\right) - \tilde{Q}_i^{\ *}\left(s,a\right) \right| = |v\left(s,a\right) - v^*\left(s,a\right)| < \varepsilon, \ \forall s \in \tilde{\mathcal{S}} \tag{7}$$

We restate the theorem assumption that the attention weights are $\delta$-close for up to $M$ objects:

$$\left| \frac{\alpha\left(s_i,s_j,a\right)}{\sum_{l=1}^{N} \alpha\left(s_i,s_l,a\right)} - \frac{\alpha^*\left(s_i,s_j,a\right)}{\sum_{l=1}^{N} \alpha^*\left(s_i,s_l,a\right)} \right| \le \delta, \quad \forall j \in 1,...,N, \quad \forall N \in 1,...,M. \tag{8}$$

Using equation 8 and the fact that $\alpha(\cdot), \alpha^*(\cdot) \in \mathbb{R}^+$, from Lemma 5 we obtain that $\forall k \in [1, N-1]$:

$$\left| \frac{\sum_{j=1}^N \alpha(s_i, s_j, a)}{\sum_{l=1}^{N+k} \alpha(s_i, s_l, a)} - \frac{\sum_{j=1}^N \alpha^*(s_i, s_j, a)}{\sum_{l=1}^{N+k} \alpha^*(s_i, s_l, a)} \right| = \left| \sum_{j=1}^N \frac{\alpha(s_i, s_j, a)}{\sum_{l=1}^{N+k} \alpha(s_i, s_l, a)} - \frac{\alpha^*(s_i, s_j, a)}{\sum_{l=1}^{N+k} \alpha^*(s_i, s_l, a)} \right| \leq$$

$$\leq \sum_{j=1}^N \left| \frac{\alpha(s_i, s_j, a)}{\sum_{l=1}^{N+k} \alpha(s_i, s_l, a)} - \frac{\alpha^*(s_i, s_j, a)}{\sum_{l=1}^{N+k} \alpha^*(s_i, s_l, a)} \right| \leq \sum_{j=1}^N 2\delta = 2N\delta \Rightarrow$$

$$\Rightarrow \left| \frac{\sum_{j=1}^N \alpha(s_i, s_j, a)}{\sum_{l=1}^{N+k} \alpha(s_i, s_l, a)} - \frac{\sum_{j=1}^N \alpha^*(s_i, s_j, a)}{\sum_{l=1}^{N+k} \alpha^*(s_i, s_l, a)} \right| \leq 2N\delta, \ \forall i \in [1, N] \tag{9}$$

Using equation 5, equation 7 and equation 8, from Lemma 7 we have:

$$\left| \tilde{Q}_i(s_1, ..., s_N, a) - \tilde{Q}_i^*(s_1, ..., s_N, a) \right| \leq \varepsilon + \frac{N}{1-\gamma}\delta, \ \forall i \in [1, N] \tag{10}$$

Note that we can decompose $\tilde{Q}_i$ in the following manner:

$$\tilde{Q}_i(s_1, ..., s_{M+k}, a) = \frac{1}{\sum_{j=1}^{M+k} \alpha(s_i, s_j, a)} \sum_{j=1}^{M+k} [\alpha(s_i, s_j, a) v(s_j, a)] =$$

$$= \frac{1}{\sum_{j=1}^{M+k} \alpha(s_i, s_j, a)} \sum_{j=1}^M [\alpha(s_i, s_j, a) v(s_j, a)] + \frac{1}{\sum_{j=1}^{M+k} \alpha(s_i, s_j, a)} \sum_{j=M+1}^{M+k} [\alpha(s_i, s_j, a) v(s_j, a)] =$$

$$= \frac{\sum_{j=1}^M \alpha(s_i, s_j, a)}{\sum_{j=1}^{M+k} \alpha(s_i, s_j, a)} \tilde{Q}_i(s_1, ..., s_M, a) + \frac{\sum_{j=M+1}^{M+k} \alpha(s_i, s_j, a)}{\sum_{j=1}^{M+k} \alpha(s_i, s_j, a)} \tilde{Q}_i(s_i, s_{M+1}, ..., s_{M+k}, a) +$$

$$- \frac{\alpha(s_i, s_i, a)}{\sum_{j=1}^{M+k} \alpha(s_i, s_j, a)} v(s_i, a)$$

Using this decomposition for $\tilde{Q}_i$ and $\tilde{Q}_i^*$:

$$\left| \tilde{Q}_i(s_1, ..., s_{M+k}, a|s_i) - \tilde{Q}_i^*(s_1, ..., s_{M+k}, a) \right| \leq$$

$$\leq (\#) \left| \frac{\sum_{j=1}^M \alpha(s_i, s_j, a)}{\sum_{j=1}^{M+k} \alpha(s_i, s_j, a)} \tilde{Q}_i(s_1, ..., s_M, a) - \frac{\sum_{j=1}^M \alpha^*(s_i, s_j, a)}{\sum_{j=1}^{M+k} \alpha^*(s_i, s_j, a)} \tilde{Q}_i^*(s_1, ..., s_M, a) \right| +$$

$$(\#\#) \left| \frac{\sum_{j=M+1}^{M+k} \alpha(s_i, s_j, a)}{\sum_{j=1}^{M+k} \alpha(s_i, s_j, a)} \tilde{Q}_i(s_i, s_{M+1}, ..., s_{M+k}, a) - \frac{\sum_{j=M+1}^{M+k} \alpha^*(s_i, s_j, a)}{\sum_{j=1}^{M+k} \alpha^*(s_i, s_j, a)} \tilde{Q}_i^*(s_i, s_{M+1}, ..., s_{M+k}, a) \right| +$$

$$(\#\#\#) \left| \frac{\alpha(s_i, s_i, a)}{\sum_{j=1}^{M+k} \alpha(s_i, s_j, a)} v(s_i, a) - \frac{\alpha^*(s_i, s_i, a)}{\sum_{j=1}^{M+k} \alpha^*(s_i, s_j, a)} v^*(s_i, a) \right|$$

Using equation 9 and equation 10 with Lemma 5 we obtain the following bound for the first term:

$$(\#) = \left| \frac{\sum_{j=1}^M \alpha(s_i, s_j, a)}{\sum_{j=1}^{M+k} \alpha(s_i, s_j, a)} \tilde{Q}_i(s_1, ..., s_M, a) - \frac{\sum_{j=1}^M \alpha^*(s_i, s_j, a)}{\sum_{j=1}^{M+k} \alpha^*(s_i, s_j, a)} \tilde{Q}_i^*(s_1, ..., s_M, a) \right| \leq$$

$$\leq \frac{\frac{\sum_{j=1}^M \alpha(s_i, s_j, a)}{\sum_{j=1}^{M+k} \alpha(s_i, s_j, a)} + \frac{\sum_{j=1}^M \alpha^*(s_i, s_j, a)}{\sum_{j=1}^{M+k} \alpha^*(s_i, s_j, a)}}{2} \left( \varepsilon + \frac{M}{1-\gamma}\delta \right) + \frac{\tilde{Q}_i(s_1, ..., s_M, a) + \tilde{Q}_i^*(s_1, ..., s_M, a)}{2} 2M\delta \leq$$

$$\leq \varepsilon + \frac{M}{1-\gamma}\delta + \frac{2M}{1-\gamma}\delta$$

Similarly for the other two terms we obtain:

$$(\#\#) \leq \varepsilon + \frac{k}{1-\gamma}\delta + \frac{2k}{1-\gamma}\delta$$

$$(\#\#\#) \leq \varepsilon + \frac{2}{1-\gamma}\delta$$

Putting the three terms together we have:

$$\left| \tilde{Q}_i\left(s_1, ..., s_{M+k}, a\right) - \tilde{Q}_i^*\left(s_1, ..., s_{M+k}, a|\right) \right| \leq 3\varepsilon + \frac{3\left(M+k\right)+2}{1-\gamma}\delta$$

The same result is obtained for $M < i \leq M + k$ by similarly repeating derivations above, starting from the decomposition of $\tilde{Q}_i$.

Using this final result we obtain our desired upper bound:

$$\left| \hat{Q}\left(s_1, ..., s_{M+k}, a\right) - Q^*\left(s_1, ..., s_{M+k}, a\right) \right| \leq$$

$$\leq \frac{1}{M+k} \sum_{i=1}^{M+k} \left| \tilde{Q}_i\left(s_1, ..., s_{M+k}, a\right) - \tilde{Q}_i^*\left(s_1, ..., s_{M+k}, a\right) \right| \leq$$

$$\leq 3\varepsilon + \frac{3\left(M+k\right)+2}{1-\gamma}\delta, \ \ \forall k \in [1, M-1]$$

$\square$

### F.3 RELAXING THE ASSUMPTION ON THE ATTENTION WEIGHTS

Theorem 2 does not exactly fit Definition 3 because it makes an assumption on the difference between the optimal and approximated attention weights (equation 8). In this section we present modifications to the assumptions of the theorem that allow us to alleviate this assumption, and bound the difference as a function of the Q-value approximation error $\varepsilon$ alone.

The first additional assumption we make on the state space $\mathcal{S}^N$ is that individual object states are *distinguishable*, and separated by some constant $C$:

**Assumption 8.** $\mathcal{S}^N = \{\{s_i\}_{i=1}^N, \quad s_i \in \tilde{\mathcal{S}} \quad | \quad \forall s_j, s_k, \quad \|s_j - s_k\|_1 \geq C > 0\}$.

The second assumption we add is that the attention-value function $v^*(\cdot)$ is object-distinguishing, i.e. $\forall \|s_i - s_j\|_1 \geq C > 0 \rightarrow |v^*(s_i, a) - v^*(s_j, a)| \geq \lambda > 0$:

**Assumption 9.** $\forall S \in \mathcal{S}^N, \forall a \in \mathcal{A}, \ \forall N \in \mathbb{N}$ we have:
$Q^*(s_1, ..., s_N, a) = \frac{1}{N}\sum_{i=1}^N \tilde{Q}_i^*(s_1, ..., s_N, a)$, where
$\tilde{Q}_i^*(s_1, ..., s_N, a) = \frac{1}{\sum_{j=1}^N \alpha^*(s_i, s_j, a)} \sum_{j=1}^N \alpha^*(s_i, s_j, a) v^*(s_j, a), \alpha^*(\cdot) \in \mathbb{R}^+, v^* \in \mathbb{R}$ and satisfies $|v^*(s_i, a) - v^*(s_j, a)| \geq \lambda > 0$.

We state our compositional generalization result under these assumptions as follows:

**Theorem 10.** *Let Assumptions 8 and 9 hold. Let $\hat{Q}$ be an approximation of $Q^*$ with an identical structure. Assume that $\forall s \in \mathcal{S}^N, \forall a \in \mathcal{A}, \ \forall N \in [1, M]$ we have $\left| \hat{Q}(s_1, ..., s_N, a) - Q^*(s_1, ..., s_N, a) \right| < \varepsilon$. Then $\forall s \in \mathcal{S}^{M+k}, \forall a \in \mathcal{A}, \forall k \in [1, M-1]$:*

$$\left| \hat{Q}(s_1, ..., s_{M+k}, a) - Q^*(s_1, ..., s_{M+k}, a) \right| \leq \left( \frac{12(M+k)}{\lambda(1-\gamma)} + \frac{8}{\lambda(1-\gamma)} + 3 \right)\varepsilon,$$

*where $\lambda > 0$ is a constant independent of $\varepsilon$.*

This theorem thus states that the class of functions described in Assumption 9 admits compositional generalization on $\mathcal{S}$ (defined in Assumption 8), as defined by Definition 3.

We now prove Theorem 10.

We start by showing that for the self-attention class of functions, if two functions are $\varepsilon$-similar for any set of objects that are $\lambda$-different, it must mean that the attention weights are also similar.

**Lemma 11.** *Consider the functions $v, v^* : \tilde{\mathcal{S}} \to \mathbb{R}^+$ and $\alpha, \alpha^* : \tilde{\mathcal{S}} \to \mathbb{R}^+$. Assume that for any $N \in 1, \ldots, M$, and for any $\{s_i\}_{i=1}^N \in \mathcal{S}^N$ it holds that:*

$$|v(s_j) - v(s_k)| \geq \lambda \quad \forall j \neq k \in 1, \ldots, N, \quad \lambda > 0, \tag{11}$$

*and*

$$\left| \frac{\sum_{i=1}^N \alpha(s_i)v(s_i)}{\sum_{i=1}^N \alpha(s_i)} - \frac{\sum_{i=1}^N \alpha^*(s_i)v^*(s_i)}{\sum_{i=1}^N \alpha^*(s_i)} \right| \leq \varepsilon, \quad \varepsilon > 0, \tag{12}$$

*Then*

$$\left| \frac{\alpha(s_i)}{\sum_{j=1}^N \alpha(s_j)} - \frac{\alpha^*(s_i)}{\sum_{j=1}^N \alpha^*(s_j)} \right| \leq \frac{4\varepsilon}{\lambda}, \quad \forall i \in 1, \ldots, N, \quad \forall N \in 1, \ldots, M.$$

*Proof.* From the case of $N = 1$ of equation 12 we have that for every $s \in \tilde{\mathcal{S}}$:

$$|v(s) - v^*(s)| \leq \varepsilon. \tag{13}$$

Consider the case of $N = 2$. Let $\bar{\alpha} = \frac{\alpha(s_1)}{\sum_{i=1}^2 \alpha(s_i)}$, and similarly $\bar{\alpha}^* = \frac{\alpha^*(s_1)}{\sum_{i=1}^2 \alpha^*(s_i)}$. We have:

$$\begin{aligned}
&|\bar{\alpha}v(s_1) + (1 - \bar{\alpha})v(s_2) - (\bar{\alpha}^* v(s_1) + (1 - \bar{\alpha}^*)v(s_2))| \\
&= |\bar{\alpha}v(s_1) + (1 - \bar{\alpha})v(s_2) - (\bar{\alpha}^* v^*(s_1) + (1 - \bar{\alpha}^*)v^*(s_2)) + \bar{\alpha}^*(v^*(s_1) - v(s_1)) + (1 - \bar{\alpha}^*)(v^*(s_2) - v(s_2))| \\
&\leq |\bar{\alpha}v(s_1) + (1 - \bar{\alpha})v(s_2) - (\bar{\alpha}^* v^*(s_1) + (1 - \bar{\alpha}^*)v^*(s_2))| + \bar{\alpha}^* \varepsilon + (1 - \bar{\alpha}^*)\varepsilon \leq \varepsilon + \varepsilon = 2\varepsilon
\end{aligned} \tag{14}$$

where the first inequality is from equation 13 and the second inequality is from equation 12. Manipulating the terms above we obtain:

$$|(\bar{\alpha} - \bar{\alpha}^*)(v(s_1) - v(s_2))| \leq 2\varepsilon. \tag{15}$$

Now, from equation 11 and equation 15 we get that for any $s_1, s_2 \in \mathcal{S}^2$:

$$\left| \frac{\alpha(s_i)}{\sum_{j=1}^2 \alpha(s_j)} - \frac{\alpha^*(s_i)}{\sum_{j=1}^2 \alpha^*(s_j)} \right| \leq \frac{2\varepsilon}{\lambda} < \frac{4\varepsilon}{\lambda}, \quad i = 1, 2.$$

The claim holds for $N = 2$.

Let us now consider the case of $N = 3$. Assume without loss of generality that $v(s_1) < v(s_2) < v(s_3)$. Denote $\gamma_i = \frac{\alpha(s_i)}{\sum_{j=1}^3 \alpha(s_j)}$, $\hat{v}_1 = v(s_1)$ and $\hat{v}_{2,3} = \frac{\gamma_2 v(s_2)}{\gamma_2 + \gamma_3} + \frac{\gamma_3 v(s_3)}{\gamma_2 + \gamma_3}$.
We rewrite the weighted sum using this notation:

$$\begin{aligned}
\frac{\sum_{i=1}^3 \alpha(s_i)v(s_i)}{\sum_{i=1}^3 \alpha(s_i)} &= \gamma_1 v(s_1) + \gamma_2 v(s_2) + \gamma_3 v(s_3) = \\
&= \gamma_1 v(s_1) + (\gamma_2 + \gamma_3)\left( \frac{\gamma_2 v(s_2)}{\gamma_2 + \gamma_3} + \frac{\gamma_3 v(s_3)}{\gamma_2 + \gamma_3} \right) = \gamma_1 \hat{v}_1 + (1 - \gamma_1)\hat{v}_{2,3}
\end{aligned}$$

Then we can rewrite equation 12 for this case in the following manner:

$$\left| \gamma_1 \hat{v}_1 + (1 - \gamma_1)\hat{v}_{2,3} - \left( \gamma_1^* \hat{v}_1^* + (1 - \gamma_1^*)\hat{v}_{2,3}^* \right) \right| \leq \varepsilon.$$

Since $\hat{v}_{2,3}$ is a weighted average of $v(s_2)$ and $v(s_3)$, we can say that $|\hat{v}_1 - \hat{v}_{2,3}| \geq \lambda$. Additionally, from the case of $N = 2$ of equation 12 we have that $|\hat{v}_{2,3} - \hat{v}_{2,3}^*| \leq \varepsilon$. Using the above with a similar derivation of equation 14 we obtain for $s_1$:

$$\left| \frac{\alpha(s_1)}{\sum_{j=1}^3 \alpha(s_j)} - \frac{\alpha^*(s_1)}{\sum_{j=1}^3 \alpha^*(s_j)} \right| \leq \frac{2\varepsilon}{\lambda},$$

and

$$\left| \frac{\alpha(s_2) + \alpha(s_3)}{\sum_{j=1}^{3} \alpha(s_j)} - \frac{\alpha^*(s_2) + \alpha^*(s_3)}{\sum_{j=1}^{3} \alpha^*(s_j)} \right| \leq \frac{2\varepsilon}{\lambda}. \tag{16}$$

Symmetrically we can obtain the following bound for $s_3$:

$$\left| \frac{\alpha(s_3)}{\sum_{j=1}^{3} \alpha(s_j)} - \frac{\alpha^*(s_3)}{\sum_{j=1}^{3} \alpha^*(s_j)} \right| \leq \frac{2\varepsilon}{\lambda}, \tag{17}$$

and we use equation 16 and equation 17 to obtain a bound for $s_2$:

$$\left| \frac{\alpha(s_2)}{\sum_{j=1}^{3} \alpha(s_j)} - \frac{\alpha^*(s_2)}{\sum_{j=1}^{3} \alpha^*(s_j)} \right| =$$

$$= \left| \frac{\alpha(s_2) + \alpha(s_3) - \alpha(s_3)}{\sum_{j=1}^{3} \alpha(s_j)} - \frac{\alpha^*(s_2) + \alpha^*(s_3) - \alpha^*(s_3)}{\sum_{j=1}^{3} \alpha^*(s_j)} \right| \leq$$

$$\leq \left| \frac{\alpha(s_2) + \alpha(s_3)}{\sum_{j=1}^{3} \alpha(s_j)} - \frac{\alpha^*(s_2) + \alpha^*(s_3)}{\sum_{j=1}^{3} \alpha^*(s_j)} \right| + \left| \frac{\alpha(s_3)}{\sum_{j=1}^{3} \alpha(s_j)} - \frac{\alpha^*(s_3)}{\sum_{j=1}^{3} \alpha^*(s_j)} \right| \leq$$

$$\leq \frac{2\varepsilon}{\lambda} + \frac{2\varepsilon}{\lambda} = \frac{4\varepsilon}{\lambda}.$$

The claim holds for $N = 3$.

Now consider the case of $3 < N \leq M$. Assume without loss of generality that $v(s_1) < v(s_2) < \cdots < v(s_N)$. Denote the normalized attention weights $\gamma_i = \frac{\alpha(s_i)}{\sum_{j=1}^{N} \alpha(s_j)}$ and the partial weighted sum $\hat{v}_{k,\dots,l} = \sum_{j=k}^{l} \frac{\gamma_j v(s_j)}{\sum_{j=k}^{l} \gamma_j}$. Given $i \in 2, \dots, N-1$, we can regroup the values such that we have $\hat{v}_{1,\dots,i-1} < \hat{v}_i < \hat{v}_{i+1,\dots,N}$ and corresponding weights $\sum_{j=1}^{i-1} \gamma_j$, $\gamma_i$, $\sum_{j=i+1}^{N} \gamma_j$. Notice that by definition, $\hat{v}_i = v_i$.

The new set of values $\{\hat{v}_{1,\dots,i-1}, \hat{v}_i, \hat{v}_{i+1,\dots,N}\}$ are $\lambda$-far from each other since $\hat{v}_{1,\dots,i-1}$ and $\hat{v}_{i+1,\dots,N}$ are weighted averages of values, $\{v_1, \dots, v_{i-1}\}$ and $\{v_{i+1}, \dots, v_N\}$, that are $\lambda$-far from $\hat{v}_i = v_i$.

From equation 12 we have that each $\hat{v}$ is $\varepsilon$-close to its $\hat{v}^*$ counterpart. To see why this is true, notice that $\gamma_i$ is a normalized $\alpha_i$ and therefore:

$$\hat{v}_{k,\dots,l} = \sum_{j=k}^{l} \frac{\gamma_j v(s_j)}{\sum_{j=k}^{l} \gamma_j} = \sum_{j=k}^{l} \frac{\frac{\alpha(s_j)}{\sum_{m=1}^{N} \alpha(s_m)} v(s_j)}{\sum_{j=k}^{l} \frac{\alpha(s_j)}{\sum_{m=1}^{N} \alpha(s_m)}} = \sum_{j=k}^{l} \frac{\alpha(s_j) v(s_j)}{\sum_{j=k}^{l} \alpha(s_j)} = \frac{\sum_{j=k}^{l} \alpha(s_j) v(s_j)}{\sum_{j=k}^{l} \alpha(s_j)}$$

So we have:

$$\left| \hat{v}_{k,\dots,l} - \hat{v}_{k,\dots,l}^* \right| = \left| \frac{\sum_{j=k}^{l} \alpha(s_j) v(s_j)}{\sum_{j=k}^{l} \alpha(s_j)} - \frac{\sum_{j=k}^{l} \alpha^*(s_j) v^*(s_j)}{\sum_{j=k}^{l} \alpha^*(s_j)} \right| \leq \varepsilon$$

We therefore return to the case of $N = 3$ for values $\{\hat{v}_{1,\dots,i-1}, \hat{v}_i, \hat{v}_{i+1,\dots,N}\}$ and weights $\{\sum_{j=1}^{i-1} \gamma_j, \gamma_i, \sum_{j=i+1}^{N} \gamma_j\}$ and have that $|\gamma_i - \gamma_i^*| = \left| \frac{\alpha(s_i)}{\sum_{j=1}^{N} \alpha(s_j)} - \frac{\alpha^*(s_i)}{\sum_{j=1}^{N} \alpha^*(s_j)} \right| \leq \frac{4\varepsilon}{\lambda}$. This applies for every $i \in 2, \dots, N-1$.

The bound on $|\gamma_1 - \gamma_1^*|$ and $|\gamma_N - \gamma_N^*|$ is obtained from the case of $i = 2$ and $i = N - 1$ since the attention weights in these cases can be divided to $\{\gamma_1, \gamma_2, \sum_{j=3}^{N} \gamma_j\}$ and $\{\sum_{j=1}^{N-2} \gamma_j, \gamma_{N-1}, \gamma_N\}$ respectively.

And thus, we obtain the desired bound for all $N \in 1, \dots, M$.

$\square$

The proof of Theorem 10 follows by substituting equation 8 with the following bound on the normalized attention weights:

From Assumption 9:

$$|v(s_j, a) - v(s_k, a)| \geq \lambda \quad \forall j \neq k \in 1, \ldots, N. \tag{18}$$

Using Assumption 9, the $\varepsilon$-optimality assumption and equation 18 we can bound the difference between the sets of approximated and optimal normalized attention weights with Lemma 11 and obtain:

$$\left| \frac{\alpha(s_i, s_j, a)}{\sum_{l=1}^{N} \alpha(s_i, s_l, a)} - \frac{\alpha^*(s_i, s_j, a)}{\sum_{l=1}^{N} \alpha^*(s_i, s_l, a)} \right| \leq \frac{4\varepsilon}{\lambda}, \quad \forall j \in 1, \ldots, N, \quad \forall N \in 1, \ldots, M. \tag{19}$$

We substitute the bound of $\delta$ with $\frac{4\varepsilon}{\lambda}$ in the derivation and obtain the desired bound:

$$\left| \hat{Q}(s_1, \ldots, s_{M+k}, a) - Q^*(s_1, \ldots, s_{M+k}, a) \right| \leq \left( \frac{12(M+k)}{\lambda(1-\gamma)} + \frac{8}{\lambda(1-\gamma)} + 3 \right) \varepsilon$$

