# OpenReview forum: "Entity-Centric Reinforcement Learning for Object Manipulation from Pixels"
_ICLR.cc/2024/Conference — ICLR 2024 spotlight_

### Official Review · Reviewer_aSoT · 2023-10-27

**Soundness:** 3 good
**Presentation:** 3 good
**Contribution:** 3 good
**Rating:** 8
**Confidence:** 4

**Summary:**

This paper presents an object- or entity-centric RL algorithm for learning goal-conditioned manipulation. As object-centric representations, the authors use the Deep Latent Particles (DLP) method. The novelty is in the policy and Q-network, for which they propose an Entity interaction transformer (EIT), which is a transformer-based architecture to process the structured per-object latent representations. They test the method in an object manipulation task, with a robot manipulator and 2 static viewpoints provided as observations. They adopt a goal-conditioned RL setup, where the goal state is provided as target images, and introduce a Chamfer reward term to train the policy and Q function. The experiments show that their method can match the performance of another structured latent state method (SMORL), and outperform it when using image goals. Moreover, they demonstrate compositional generalization, where an agent trained on i.e. 3 colored cubes can generalize to a task with N colored cubes.

**Strengths:**

- This paper provides various novel contributions such as the transformer architecture for the Q and policy network, the Champfer reward to train policies conditioned on goal images, and demonstrates compositional generalization.

- The experimental results show ablations for the various components, such as the Champfer reward, using object-centric structured latent state spaces and using multiple views.

**Weaknesses:**

- The method seems very tied to the experimental setup of having a robot manipulator that needs to push objects to a particular location. Some of the proposed novelties such as the Champfer reward don't seem very applicable beyond this use case.

- The experiments are limited to a single environment of colored cubes. It would be interesting to see whether the approach can scale to various objects (for example YCB objects), and more cluttered scenes.

- As hinted by the authors SMORL is more sample efficient, as it learns to manipulate a single object and can then generalize to the others. This seems to be an essential feature / reason to go to object-centric approaches.

**Questions:**

- An important rationale for object-centric representations for RL is that once you learn a policy on one object, you can apply it to other objects (i.e. explaining the sample efficiency gap with SMORL). Why did the authors choose to discard this feature in their architecture, and would there be options to combine the strengths of both?

P.S: Fig 6 caption has a typo "mathcing"

---

> ### Author Response · Authors · 2023-11-19
>
> Thank you for acknowledging the novelty and contribution of our work.
>
> **Experimental Setup**
>
> *Focus on Pushing Tasks* -  We have added experiments with a more complex task of orienting a T-shaped object. We believe that various other manipulation tasks could easily be handled by our policy, given a proper reward function during training.
>
> *Chamfer Reward* - Designing a reward function from images is a difficult task, and while we make some progress on it using the Chamfer reward (which can also be extended to the orientation task), we acknowledge that it is not yet solved.
>
> *Objects in the Environment* - The experiments we have added in the new revision may shed light on some of your concerns regarding the types of objects our method can deal with. The push-T task demonstrates manipulation of an object that is not a cube with more complex dynamics (see Appendix B.4 and “Push-T” tab on our website). In addition, we present zero-shot generalization results on variations of object properties, including cuboids of different dimensions (results are presented in Table 6 and discussed in Appendix B.2  in the new revision). While zero-shot generalization to objects with different properties is partial on some scenarios, the success rates are not negligible and therefore hint at potential for few-shot generalization. These results also imply that our method can handle training with different types of objects to begin with.
>
> *Cluttered Scenes* - The Group-Push experiment shows the ability of our agent to deal (zero-shot) with a large amount of objects densely occupying the table (see our website for demonstrations).
>
> **Sample Efficiency**
>
> *Comparison to SMORL* -  We would like to emphasize that SMORL’s policy does not generalize to multiple cubes - it is trained to perform single-cube manipulation and does so also during inference. A scripted meta-policy is used to cycle between the objects in the environment and solve the sub-goals one by one during inference, disregarding the other objects while doing so. While SMORL is more sample efficient in the sense that it only requires learning single-object manipulation, it is fundamentally limited in performing complex multi-object tasks that require modeling interactions, as we demonstrate in our experiments (see Figure 4d, 4f and Table 1).
>
> *Best of Both Approaches* - We designed our method to learn simultaneous multi-object manipulation because it is applicable to a larger group of tasks than learning on a single object is, as we demonstrate in our experiments (see Figure 4d, 4f and Table 1). We present types of sub-goals that can’t be achieved without taking all sub-goals into account (e.g. Ordered-Push).
>
> Compared to SMORL, there is indeed a tradeoff between sample efficiency and performance that originates from simplification of the learning problem. In some cases, which we claim to be closer to real-world scenarios, oversimplification significantly hurts performance.
>
> Regarding combining the strengths of both, this could be an interesting direction for future work. One way to do this would be to create a curriculum of an increasing number of objects similar to Li et al. [1]
>
> [1] Richard Li, Allan Jabri, Trevor Darrell, and Pulkit Agrawal. Towards practical multi-object manipulation using relational reinforcement learning. In 2020 ieee international conference on robotics and automation (icra), pp. 4051–4058. IEEE, 2020.

---

> > ### Comment · Reviewer_aSoT · 2023-11-22
> >
> > I'd like to thank the authors for their responses, as well as for adding the extra experiments that touch upon the points I made. I also increased my score accordingly.

---

### Official Review · Reviewer_PeH5 · 2023-10-30

**Soundness:** 3 good
**Presentation:** 2 fair
**Contribution:** 3 good
**Rating:** 6
**Confidence:** 4

**Summary:**

Authors solve table-top goal conditioned tasks form pixels using particles encoding and a transformer based RL. This is an interesting improvement over previous SOTA works and its major weakness are: 1) entities are fixed cubes-with-specific-color, thus there is no possibility to generalize to other objects with different properties, 2) interaction between the objects has not properly been demonstrated and 3) related work could be improved.

**Strengths:**

-	The paper is clear, well written and the topic is very interesting for the community
-	The viewpoint is a nice work around to solve depth ambiguities
-	The conditional goal transformed is sound and nicely implemented for an actor-critic RL.
-	Inputs are just pixels, thus making the problem very complex.
-	Clever use of the Chamfer Distance with the particles.
-	Examples animations provided that shows the system working.

**Weaknesses:**

**Related work**

The literature review is too focused on RL and could be improved.

Example of missing SOTA object-centric perception:
M. Traub et al. Learning what and where: Disentangling location and identity tracking without supervision,”  International Conference on Learning Representations, 2023.

“learning to manipulate” The citations related to manipulation are only for image segmentation, there is a scarce but very good literature on object-centric manipulation. The majority with full observability but some from pixels.

- Works based on Interaction Networks, Propagation Networks and Graph networks. E.g., A. Sanchez-Gonzalez, N. Heess, J. T. Springenberg, J. Merel, M. Riedmiller, R. Hadsell, and P. Battaglia. Graph networks as learnable physics engines for inference and control.

- Examples from pixels:
van Bergen & Lanillos (2022). Object-based active inference. Workshop on Causal Representation Learning @ UAI 2022.
Driess et al. "Learning multi-object dynamics with compositional neural radiance fields." Conference on Robot Learning. PMLR, 2023.

- Finally, regarding the use of particles for robotic control I really think that this work is seminal:
Levine, S. et al. (2016). End-to-end training of deep visuomotor policies. The Journal of Machine Learning Research, 17(1), 1334-1373.

As an aside comment, we can find similar table-top behaviours as the one presented here using LLMs, e.g., “Palm-e: An embodied multimodal language model.”

**Methods:**

This sentence requires elaboration: “Obviously, we cannot expect compositional generalization on multi-object tasks which are different for every N.”

Assumption 1. Probably you are using a standard notation but please explain what is \alpha* and v*.

Could you explain the consequence of Theorem 2.

Why did you use off-policy algorithm TD3?

¿Why do you need RL to train the DLP? It was mentioned that this module is pretrained, so no goal would be needed. Otherwise, you constraint the training for the defined goals that are set by the designer.

Goal definition – Using the encoder. This is a common technique but prevents for proper generalization. How you would encode in this architecture non-predefined goals?, like move red objects to the left.

Particles and only cubes. Using particles is very interesting, but evaluation with non-cube objects is not tested. This means that it could be that the experiments are assuming that the objects are point-mass entities. This would prevent generalization. In particular, the definition of cube-red as a single entity seems very restricted so you cannot perform behavioural operations on other shapes with different colours or other properties.

Also this rises the problem of permutation invariant, maintaining the identity of an object may is important in tasks that object permanence is needed for instance in dynamic-sequential tasks.

**Experiments**

¿Why adjacent goals require interactions? This can be solved reactively.

I find very interesting the Ordered-Push. Should be the EIT trained for each task or it is trained on all tasks and the executed?

I understand that you relegated the Chamfer distance to the Appendix, but it could be great that at least a written explanation is placed (or the equation) to understand how the rewards works.

Using this distance (and the L2)  as rewards why is RL needed, would it be enough to use a KL as objective function? Or are there other rewards used?

What is state input? Full observability?

The agent is learning arm-object interaction thanks to the RL approach but it is not clear that the system is learning objects interaction.
Compositional generalization. While I agree that training on N objects and then executing the task with less and more objects shows generalization capabilities. This does not necessarily endorses composition.

Could you explain how the system changes when including more objects at the level of the  DLP and the EIT?

Baselines: The text says: “We use DLP as the pre-trained OCR for this method for a fair comparison”, but then SMORL is only compared in the results showed with “state” access. Does this mean that this is without using pixels as input.

It is interesting that using RL also unstructured approach cannot handle the complexity. We obtained similar results using an ELBO loss. However, this makes the comparison too naïve. As the comparison of your algorithm is against full observable (state) and unstructured.

**Minor comments**

- Please check open quotes, in latex you can use ``word”
- Self attention -> Self-attention

**Questions:**

See weaknesses.

---

> ### Author Response · Authors · 2023-11-19
>
> Thank you for acknowledging the complexity of the tasks we deal with and the strengths of different aspects of our method. We also thank you for your questions and interest in our work.
>
> **Related Work**
>
> Thank you for pointing out the various papers to us, widening the scope of our related work to both model-based planning and active inference methods. We have added them to the related work section of the new revision.
>
> Regarding Palm-e, it is an LLM-based planning algorithm that assumes access to a low-level manipulation policy. Here we use RL to learn a single policy that solves the entire task, both single-object manipulation and planning to achieve the overall goal while accounting for object-object interaction.
>
> **Method**
>
> *”This sentence requires elaboration: “Obviously, we cannot expect compositional generalization on multi-object tasks which are different for every N.””*
>
> We expect compositional generalization to an increasing number of objects where the objects are similar to the ones seen during training and the basic task remains the same with the increased number of objects, such as pushing each object to a desired goal location.
>
> *”Assumption 1. Probably you are using a standard notation but please explain what is \alpha* and v*.”*
>
> We reference the supplementary background for the attention mechanism (Appendix E) where $\alpha$ represents the attention weights (produced from keys and queries) and $v$ represents the values. * denotes the optimal versions of those functions (per our assumption that the optimal Q function has an attention structure).
>
> *”Could you explain the consequence of Theorem 2.”*
>
> The theorem bounds the approximation error of a learned Q-function vs. the optimal Q-function, when both have a self-attention structure. The Q-function is learned on environments with $M$ objects or less, and is epsilon-close to the optimal Q-function in this case (less than $M$ objects). The optimal Q-function is optimal for all $N$. We show that with no additional training, the approximation error increases linearly in the number of objects. This is a notion of compositional generalization defined through the Q-function, and we show that a Q-function with a self-attention structure obtains it.
>
> What this implies is that the Q-function used to optimize a policy on $M$ objects is not very far from the optimal Q-function for $M+k$ objects and thus the same policy should obtain similar returns even when deployed on environments with an increasing number of objects.
>
> *”Why did you use off-policy algorithm TD3?”*
>
> TD3 is a SOTA off-policy model free deep RL algorithm based on DDPG which has been used in similar tasks in previous work, including the original HER paper. It is well suited for goal-conditioned RL because it allows dynamically relabeling goals in the replay buffer transitions sampled during training, thereby improving sample efficiency.
>
> *”¿Why do you need RL to train the DLP? It was mentioned that this module is pretrained, so no goal would be needed. Otherwise, you constraint the training for the defined goals that are set by the designer.”*
>
> We do not need RL to train the DLP nor do we claim this. The DLP is pretrained on image data collected by a random policy. This is not related to the goals. Goals in our experimental setup are defined as images of the scene with a specific object configuration.
>
> *”Goal definition – Using the encoder. This is a common technique but prevents for proper generalization. How you would encode in this architecture non-predefined goals?, like move red objects to the left.”*
>
> What are you referring to by predefined goals? The goals in our setting are not predefined, they are provided as an image of the desired configuration by the user. This image is converted to the latent particle representation using the pre-trained DLP encoder, which is able to encode images it has not seen during training but that come from the same distribution. Moreover, it is able to generalize to images with a larger/fewer number of objects than it has seen during training. These are demonstrated by the success of our method as well as in the generalization capabilities in the Cube-Sort scenario.
>
> If by “move red objects to the left” you are referring to goal specification through natural language, this is very possible and integrates well with our proposed method, assuming access to a reward function. The goal entities will consist of the natural language tokens specifying the goal and their relation to the DLP-extracted image entities will be learned via the cross-attention block in the EIT. Reward design might be a challenge in this case but this is not exclusive to our method. We leave such an interesting investigation to future work.

---

> > ### Author Response · Authors · 2023-11-19
> >
> > **Method - Continued**
> >
> > *”Particles and only cubes. Using particles is very interesting, but evaluation with non-cube objects is not tested. This means that it could be that the experiments are assuming that the objects are point-mass entities. This would prevent generalization. In particular, the definition of cube-red as a single entity seems very restricted so you cannot perform behavioural operations on other shapes with different colours or other properties.”*
> >
> > We have studied the generalization capabilities of our method to object properties not seen during training, please refer to Appendix B.2 of the new revision.
> >
> > We additionally provide experimental results on non-cube objects in our revision. The new task involves pushing a T-shaped block to a goal orientation. This task demonstrates that the DLP latent representation is expressive enough to infer implicit object properties (orientation from visual latent attributes) as well as the ability of the EIT to capture these properties and learn how to manipulate them to achieve desired goals.
> >
> > *”Also this rises the problem of permutation invariant, maintaining the identity of an object may is important in tasks that object permanence is needed for instance in dynamic-sequential tasks.”*
> >
> > The identity of an object is inferred by the EIT based on the objects’ latent features (produced by the DLP model) and is very important for the goal to be achieved. The ordering of the extracted particles does not change the Q-value or the desired policy, hence we use the permutation invariant architecture. If the identity of the object cannot be inferred from the image alone (e.g. 2 cubes of the same color), one cannot expect a goal that depends on a hidden identity to be achieved. If the goal does not distinguish between identical-looking objects, this should be reflected in the reward and will be learned by the agent.
> >
> > What are you referring to by dynamic-sequential tasks? We can think of some tasks that require object permanence given that objects are not visually distinguishable. These scenarios would require a non-markovian policy that takes history into account. An interesting direction for future work would be to expand the EIT to accept inputs from multiple timesteps as well as multiple viewpoints. This addition should be fairly easy to implement by adding another dimension to the input as well as temporal encoding.
> >
> > **Experiments**
> >
> > *”¿Why adjacent goals require interactions? This can be solved reactively.”*
> >
> > Adjacent-Goals can sometimes be solved reactively but there is a non-negligible probability that the goal won’t be achieved if not taking object-object interaction into account. This is supported by our empirical results, and is demonstrated in the supplementary material and our website (please see “Entity-Entity Interaction” tab on our website). In the video demonstration, we see that a policy that does not take object-object interaction into account consistently pushes objects away from their goal while attempting to bring another object to its desired subgoal. These cases happen often enough that SMORL experiences a performance drop of ~20% in success rate compared to non-adjacent goals and compared to our method on Adjacent-Goals (see Table 1).
> >
> > *”I find very interesting the Ordered-Push. Should be the EIT trained for each task or it is trained on all tasks and the executed?”*
> >
> > The EIT is trained for each task separately except for Adjacent-Goals. The reason for this is that the physical constraints of Ordered-Push and Small-Table are different and need to be learned (i.e., cubes will fall off the smaller table or cubes will run into a wall).
> >
> > *”I understand that you relegated the Chamfer distance to the Appendix, but it could be great that at least a written explanation is placed (or the equation) to understand how the rewards works.”*
> >
> > Thank you for pointing this out, we have added an explanation about the Chamfer reward in the main text.
> >
> > *”Using this distance (and the L2) as rewards why is RL needed, would it be enough to use a KL as objective function? Or are there other rewards used?”*
> >
> > Can you please elaborate what you mean by using the KL Divergence as the objective function? Objective for policy learning? KL between what two probability distributions?
> > RL is needed because while the reward might be differentiable with respect to the state, the dynamics of the environment are not. RL allows maximizing long horizon discounted rewards and model-free RL allows doing so without explicitly modeling the dynamics of the environment.
> >
> > *”What is state input? Full observability?”*
> >
> > State input is defined precisely in Appendix C.1 and is the ground truth $(x,y)$ coordinates of each object. For the factored representation, we add a one-hot vector as entity identifying features.

---

> > > ### Author Response · Authors · 2023-11-19
> > >
> > > **Experiments - Continued**
> > >
> > > *”The agent is learning arm-object interaction thanks to the RL approach but it is not clear that the system is learning objects interaction.”*
> > >
> > > Without learning object-object interaction, the agent will not be able to solve the Ordered-Push task with close to 100% success rates because in order to do so, it must understand the physical constraint of not being able to push one object through the other and therefore needs to insert the objects to the corridor in a specific order.
> > >
> > > We also clearly see the need for object-object interaction in the adjacent goals where the agent must understand that objects need to be manipulated around other objects so as to not push them away from their respective goal.
> > >
> > > *”Compositional generalization. While I agree that training on N objects and then executing the task with less and more objects shows generalization capabilities. This does not necessarily endorses composition.”*
> > >
> > > We refer to compositional generalization in our context as generalizing to a different number of objects (Section 4.4). While we acknowledge that this is not the only type of compositional generalization, this term is used in previous literature and includes our use of the term for variations in number of entities (“A survey on compositional generalization in applications”, Lin et al.).
> > >
> > > *”Could you explain how the system changes when including more objects at the level of the DLP and the EIT?”*
> > >
> > > The DLP model is trained to produce a representation with a fixed number of particles. Our design choice was to use a large number of particles, 24, much larger than the number of entities in the environment. The reason for this is that capturing the agent sometimes requires multiple particles and in addition, this lowers the chances of entities being overlooked by the model and not being represented by at least a single particle. The EIT easily handles the redundant particles as demonstrated by our method’s performance. Since we have a large amount of particles, even when increasing the number of objects from 3 to 12 (such as in the generalization experiments), all the objects are captured by the DLP model. The EIT is not limited in the number of objects it receives, although it always gets 24 particles as input. The difference is that with more objects, more particles represent distinguished objects (different attributes in different locations) and that requires the policy to generalize to unseen states.
> > >
> > > *”Baselines: The text says: “We use DLP as the pre-trained OCR for this method for a fair comparison”, but then SMORL is only compared in the results showed with “state” access. Does this mean that this is without using pixels as input.”*
> > >
> > > In the image-based SMORL we use DLP as the OCR. The comparison with SMORL in the image-based case is in the Chamfer Reward section of the Appendix A.2. In the graphs of the main text we compare both image-based and state-based methods with GT reward. Image-based SMORL is designed in a way that does not allow using it with a reward that is not image-based, therefore does not appear in those graphs.
> > >
> > > *”It is interesting that using RL also unstructured approach cannot handle the complexity. We obtained similar results using an ELBO loss. However, this makes the comparison too naïve. As the comparison of your algorithm is against full observable (state) and unstructured.”*
> > >
> > > Can the reviewer please elaborate on this point? Which comparison do you believe is too naive and why?
> > >
> > > If you are referring to the comparison between the state-based unstructured approach to our image-based method, we don’t believe this comparison to be naive. While it might be clear that structured approaches should perform better here, unstructured data from ground truth state observations still explicitly holds all the relevant attributes (and only those) to perform the task. What is mostly missing is the permutation invariance as an inductive bias. The fact that it learns slower than our method which is image-based, emphasizes the importance of this structure for object-centric tasks.

---

> > ### Comment · Reviewer_PeH5 · 2023-11-20
> > **Good achievement, new results**
> >
> > After reading the reviewers comments and the responses, most of my concerns were addressed or at least explained. I believe it is a good contribution and the new results shows that there is room for improvement when changing the objects features. Comparison against only full-observable settings is not optimal but understandable. Thus, I would increase my score.

---

### Official Review · Reviewer_SBTK · 2023-10-30

**Soundness:** 3 good
**Presentation:** 3 good
**Contribution:** 4 excellent
**Rating:** 8
**Confidence:** 4

**Summary:**

The paper presents an object-centric RL model that can learn to manipulate many objects and shows generalization capabilities. The main contribution is the combination of Deep Latent Particles (DLP) as entity-centric perception pipeline and a transformer for policy and Q function. By defining a reward based on feature closeness and geometric distance there is no matching between goal-image and current image required. The only caveat is that the objects need to be filtered (the robot needs to be removed).
The proposed method is pretty simple in comparison to prior work that considers entity-entity interactions.

**Strengths:**

- scalability to many objects
- a relatively simple method
- no explicit matching is required
- multiview
- supplementary contains important comparisons w.r.t. the reward etc.

**Weaknesses:**

- number of objects known.
- missing related work and baselines:
   - SRICS [1] is like SMOURL but dealing with object-object interactions
   - DAFT-RL[3]: also tackles the interaction problem and baselines therein
     DRAFT-RL is fairly recent, but it contains, IMO, relevant related work and further baselines, such as:
     NCS [3], STOVE [4] etc.
- supervision/filtering of entities such that only objects go into chamfer reward computation is hidden in the appendix
- only empirical results on one type of environment: I am wondering how well it would generalize to more cluttered scenes, e.g. to a kitchen environment

Details:
- Fig 5: too small font in the right subplot
- Appendix A: Chamfer rewards:
  The definition of $X_j$ and $Y_i$ after Eqn (1): what is the $i$ in the definition of $X_j$? Do I understand correctly, that it is all $x$ that have $y_j$ as their closest entity in $Y$?
  Also afterward, when you write how to obtain standard Chamfer, the $sum_j$ is somehow missing for the second fraction.
- I think some more information about the Generalized Density Aware Chamfer reward should go into the main text, and also that non-object particles are removed.
- A paper that also addresses many-object manipulation with an object-centric representation is [5] (not from images)

- citations/references are often published at conferences but listed as arXiv papers

[1] https://proceedings.mlr.press/v164/zadaianchuk22a.html
[2] https://arxiv.org/abs/2307.09205.pdf
[3] https://openreview.net/forum?id=fGG6vHp3W9W
[4] https://openreview.net/forum?id=B1e-kxSKDH
[5] https://openreview.net/forum?id=NnuYZ1el24C

**Questions:**

- how important is it that the robot is mostly white on a white background? What happens if a larger part of the robot is seen in the images? I suggest discussing this in the limitations. Also, the need to filter non-object entities. Other works would also move the robot to a particular position in the scene if part of the goal.
- what happens if the number of latent particles is higher than the number of entities?
- How do you compare to the above-mentioned baselines?

--- Post rebuttal update. My concerns were addressed. I changed my score from 5 to 8.

---

> ### Author Response · Authors · 2023-11-19
>
> Thank you for acknowledging the strengths of our work and simplicity of our method. We also thank you for your questions and  referring us to the various methods related to our work.
>
> **Related Work and Baselines**
>
> We have added the various methods you referred us to in the related work section. Our method is model-free and arguably much simpler than the methods you mentioned. Additionally, we introduce multiview inputs which are seamlessly integrated in a single model, and show zero-shot generalization to a substantially larger number of objects than previous work.
>
> In the following, we address each method in comparison to ours in detail.
>
> [SRICS](https://arxiv.org/abs/2109.04150)
>
> We find this work an interesting extension of SMORL (by the same authors) for *state inputs* which takes into account object-object interaction.
>
> *Method Complexity*: compared to our method from state inputs, it requires training a separate network to consider relationships and interactions. This network is essentially a world-model that is used during both RL training to define rewards and during inference to decide on the ordering of sub-goals to solve using the RL policy. Our method incorporates all of the above aspects in a single transformer-based model, making it more robust, easy to train and requiring less hyper-parameters.
>
> *Assumptions*: An assumption of SRICS is that when trying to achieve a given sub-goal, the agent should minimize its effect on other sub-goals. This assumption is explicitly integrated in the method through the “selectivity reward” used to train the SAC agent. This can be a restrictive assumption that could negatively impact performance. Consider for example the task of moving objects located at one side of the table to the other. The SRICS agent will move the objects one by one, although the optimal behavior with respect to timestep efficiency (and a negative distance reward) would be to move multiple objects simultaneously. Our method does not make such assumptions.
>
> *Inputs*: While we find this work related to ours, it is not a relevant baseline because it is not designed for image inputs. The authors mention the extension of this method to image-based inputs as future work, implying that this would not be a trivial extension.
>
> [Linear Relation Networks (LRN)](https://arxiv.org/abs/2201.13388)
>
> This method uses state inputs. It is similar to SMORL in the sense that it only considers relationships between objects and goals, therefore not accounting for object-object interaction. Due to the above, we believe SMORL is a better fit as a baseline which covers the essentials of the LRN method and improves over them (image-based, goal-conditioned via images).
>
> [CEE-US](https://arxiv.org/abs/2206.11403)
>
> Thank you for pointing out this paper to us. Although it is not an RL method and uses state observations, it does deal with exploration and compositional generalization in the multi-object manipulation setting. We have added it to the related work section.
>
> [NCS](https://arxiv.org/abs/2303.11373)
>
> This method presents a structured approach for solving object rearrangement tasks. They use a modification of SLATE as an object-centric representation of images. They then use it to construct a factored transition graph between clustered states where the nodes are states and the edges are actions. They use this graph to plan and execute actions on a high-level object rearrangement environment.
>
> *Method Complexity*: Generally, NCS is a complex method containing multiple manual (not learned) task specific substeps in each stage of the algorithm, increasing the number of hyper-parameters and making it less generally applicable. These include clustering states using K-Means, isolating which object moved in each transition based on the difference in state attributes, explicitly matching state and goal objects using the Hungarian algorithm (assumes a 1-to-1 match exists) and at each timestep, choosing the object with the largest distance from the goal as the next candidate to move, not taking into account goal constraints such as the presence of other objects in the goal position.
>
> *Assumptions*: This work makes several simplifying assumptions. They do not consider entity-entity interaction, do not handle occlusion (no agent in image), do not deal with low level control (assume high level actions). Our method is designed to address challenges that arise in the real world where these assumptions do not hold. We consider entity-entity interaction via the attention mechanism and handle occlusions by integrating multiple viewpoints in our EIT architecture. Additionally, our method is aimed at learning both low level and high level control, which prevents us from using NCS as a baseline in our experimental setup.

---

> ### Author Response · Authors · 2023-11-19
>
> **Related Work - Continued**
>
> [STOVE](https://arxiv.org/abs/1910.02425)
>
> STOVE is an object-centric video prediction model which also demonstrates world-modeling capabilities for solving a sequential decision making task via planning (Monte Carlo Tree Search). They extend their video prediction model to dealing with control by conditioning the predictions on actions and predicting not only the next frame but also the reward to enable reward-maximizing planning. In their evaluation, they consider visually simple 2D environments and discrete action spaces.
> It is not clear how this method would scale to complex 3D environments and continuous action spaces such as the ones we consider in our experimental setup. Additionally, expanding their model to multiview inputs is highly non-trivial.
>
> [DAFT-RL](https://arxiv.org/abs/2307.09205)
>
> This paper proposes an object-centric model-based approach, where the state is not only factored into entities but also to their individual attributes such as physical and visual properties. It does so by learning multiple separate networks which include 3 graphs. One is in charge of modeling the relationships between object attributes, the agent’s action and the reward. A second graph is in charge of modeling object-object interaction and the third is in charge of agent-object interaction and forward dynamics prediction.
>
> *Method Complexity*: The proposed method consists of several components and training stages. The stages are roughly separated into learning each of the graphs detailed above for each class of objects which is assumed to be known. Each network is learned by maximizing a distinct objective using different data. These do not include the policy network which is learned subsequently using the various model components. Compared to their method, our transformer-based model is trained to maximize a single objective and learns relationships between entity attributes as well as entity-entity interaction in an implicit manner in order to perform multi-object tasks.
>
> *Assumptions*: DAFT-RL makes several assumptions that explicitly affect their algorithm design choices. They assume a known set of object classes and that their attributes are both accessible from images and are disentangled in the latent image representation. The Push-T task we consider in our experiments is an example where this assumption does not hold, as both color and orientation are implicit in the latent visual features. In addition, the mass or friction coefficient of the objects for example, can’t be inferred from images alone. They also assume each action directly affects a single object at a time. We do not make these assumptions. The attention mechanism in the EIT along with representing the action as an entity in the Q-function network enables modeling agent interaction with multiple objects simultaneously.
>
> **Related Work Conclusion**
>
> We compare our method to what we see as the most relevant SOTA baselines: goal-conditioned model-free algorithms that are image-based. There are clear advantages to model-free over model-based methods (as is true for the other direction) and we see our work as a step forward in goal-conditioned object-centric model-free RL.
>
> **Environments**
>
> We have demonstrated that our method solves core challenges in multi-object manipulation such as occlusions, interactions between objects, a continuous action space, and compositional generalization. Our new results also demonstrate success on more geometrically complex objects (Push-T). These challenges, which were largely ignored in previous studies, are important for real world scenes such as kitchens. There may be additional challenges, but we believe our results show a clear progress towards handling such scenes.
>
> **Number of Objects**
>
> We wish to emphasize that the **number of objects is not known** and is not used explicitly in any part of the algorithm. The number of objects should generally affect the choice of the DLP model’s “number of particles” hyper-parameter. In our experimental setup, we set “number of particles”=24, which is much larger than the amount of objects/entities we consider in any of our environments. We found that choosing a relatively high upper bound is beneficial for RL training as it reduces the chance of objects in the scene not being represented by at least a single particle, and allows representing large entities (such as the robot arm) with multiple particles. In addition, this allows for zero-shot generalization to a significantly larger number of objects, as we demonstrate in the Group-Push experiment.

---

> > ### Author Response · Authors · 2023-11-19
> >
> > **Filtering of Entities in the Chamfer Reward**
> >
> > The filtering of entities for the reward is an implementation choice that is not directly connected to our method but to the specific tasks we consider in our experiments.
> >
> > Even when solving from GT simulation state, the agent is “filtered” from the reward in the sense that the reward is only defined on the objects of interest. This is the standard task and standard reward for block pushing in previous works. Additionally, previous work such as SMORL consider performance metrics that exclude the agent. While SMORL aims to remain completely self-supervised, including the arm in the reward when it is not the goal of the task could potentially negatively affect performance.
> >
> > While disregarding the agent in the reward when working with state observations is trivial, it is not trivial when working with images. Without an object-centric representation of images, this separation is not feasible because the agent and objects are not disentangled in the representation. Using an OCR not only enables this, but makes it straightforward with minimal supervision. The supervision we detail in the Appendix A.1 is very light and easy to acquire both in simulation and the real world.
> >
> > To your suggestion, we will refer to this choice in the main text.
> >
> > **Questions and Details**
> >
> > *Chamfer Reward Definition Clarification* - You understood the definition correctly. $x_i$ refers to $x \in X$ that have $y_j$ as the closest entity in $Y$. In the standard Chamfer distance, we have a single sum in each term which is over $x_i$ or $y_j$. In the DAC distance, the additional sum comes from the fact that we first take an average over all $x$ that are mapped to the same $y$, and then average over the different groups that are defined by this mapping. We have added details about the GDAC reward to the main text in the new revision.
> >
> > *“How important is it that the robot is mostly white on a white background?”*
> >
> > The fact that the robot is mostly white on a white background could potentially make it harder for the DLP model to distinguish them from each other when modeling the scene. In preliminary experiments during the development of our method, the DLP modeled the robot in the background. After decreasing the capacity of the latent background particle by setting its dimension to $1$, this no longer happened and the DLP successfully distinguished the robot from the background.
> >
> > *“What happens if a larger part of the robot is seen in the images? I suggest discussing this in the limitations. Also, the need to filter non-object entities.”*
> >
> > There are many cases in our setting where a large part of the robot is seen in the images. This can be seen on our website or in the supplementary material where the particle locations are visualized on the video - many of the particles are on the agent. We do not see this as a limitation of our method as the EIT learns which entities are relevant for solving the task.
> >
> > Filtering non-object entities in the EIT via masking in the attention mechanism slightly improves sample efficiency and requires no supervision over the input, as we use the latent transparency attribute of the DLP to do so (i.e., it is learned unsupervised as part of DLP pretraining). This is not a limitation of our method but rather a feature that is enabled via the use of attention and DLP. Additionally, it is not required for the agent to learn. When conducting experiments during the development of our method we saw that the agent achieves the same performance in slightly longer training times without the attention masking.
> >
> > *”Other works would also move the robot to a particular position in the scene if part of the goal.”*
> >
> > If the robot position is part of the goal, there would be no need to filter the robot particles in the Chamfer reward calculation. We believe an object pushing task should not include the robot position as part of the goal hence the need for filtering.
> >
> > Previous work such as RIG added the robot position as part of the goal - we see this as a limitation of their method as they can not separate the goal to individual entities due to the single-vector representation of the scene. SMORL considered the robot position as part of the goal during both training and inference, although the definition of their task is object manipulation, as highlighted by their performance metric which only considers the object distances from their desired goals. We believe that SMORL chose to include the robot as part of the goal in order to remain completely self-supervised. We see this as a limitation as well, as it may negatively affect performance. Such negative effects include counter productively moving the agent to its goal before moving the rest of the objects to their desired goal.

---

> > > ### Author Response · Authors · 2023-11-19
> > >
> > > **Questions and Details - Continued**
> > >
> > > *“What happens if the number of latent particles is higher than the number of entities?”*
> > >
> > > This is exactly the case in our setting. The number of latent particles is 24, much larger than the number of entities in the scene which is 4 (3 objects and the agent). This design choice is motivated by the fact that we prefer redundancy of particles in order to decrease the chance of entities being overlooked by the DLP model. This is emphasized in the video on our website (“In the Eyes of the Agent”) where multiple particles represent the agent and each object.
> > >
> > > In the case it is the other way around (more entities than latent particles), this would be a problem, but also a bad choice from the user’s side. This is not a limitation of the method as the number of particles is a controllable hyper-parameter.

---

> > > > ### Comment · Reviewer_SBTK · 2023-11-23
> > > >
> > > > Thank you for your detailed answer.
> > > > Most of my concerns are addressed. I like that you used 24 particles also for scenes with smaller number of objects.
> > > >
> > > > I have a different opinion on what is a limitation, or what should be stated there. I think it is important to specify which supervision and domain knowledge is required for your method to work well. Whether less assumptions would lead to suboptimal performance is a different question.
> > > >
> > > > Can you indicate whether:
> > > > - The push-T task (very nice BTW) is going to be referenced in the main text
> > > > - Fig 5 right will be updated

---

> ### Author Response · Authors · 2023-11-23
> **Thank you for your response - clarification**
>
> Thank you for your response. To your suggestion, we address the supervision required for the Chamfer reward in our setting in the main text and in more detail in Appendix A.1. We have increased the font size in Figure 5, but we can further increase it if the reviewer still finds it necessary. We will add the Push-T results and discussion to the main text in the next revision.

---

> > ### Comment · Reviewer_SBTK · 2023-11-23
> >
> > Thanks. Great work. I will increase my score.

---

### Official Review · Reviewer_sQ6n · 2023-10-31

**Soundness:** 3 good
**Presentation:** 3 good
**Contribution:** 3 good
**Rating:** 8
**Confidence:** 3

**Summary:**

This manuscript introduces an innovative approach that seamlessly integrates an object-centric model with a transformer to master structured representations crucial for goal-conditioned reinforcement learning (RL), particularly in scenarios entailing multiple objects or entities. The employed object-centric model, denoted as DLP, equips the framework with the capability to capture a structured portrayal of the environments. Concurrently, the transformer component adeptly models the dynamics of the entities and their intricate physical interactions. The clarity of the conceptual foundation is commendable, and the results showcased, particularly in the challenging realm of image-based control, are robust and hold promise. Furthermore, the paper hints at potential advancements in the field of compositional generalization. Given these strengths, I am inclined to recommend this paper for acceptance, acknowledging its significant contributions and merits. However, there are some unclear points in the current version and it would be better if the authors could provide clarification on them.

**Strengths:**

- **[General idea]** Overall, the concept presented in the paper is elegantly simple and straightforward—a notable strength, as this simplicity bodes well for better understanding and potential scalability of the framework. This is of particular importance, despite the approach essentially being a synthesis of OCR and transformer-based MBRL.

- **[Presentation]** The clarity and coherence of the presentation, spanning both the main paper and the appendix, are commendable, facilitating easy comprehension for the reader. Nevertheless, I have enumerated several recommendations in the subsequent sections to further enhance the manuscript.

- **[Experiments]** The experiments conducted using IsaacGym validate the method's efficacy, and the exploration of compositional generalization yields valuable insights. However, I have outlined several suggestions in the sections that follow, aimed at verifying some claims made in the algorithm's design.

**Weaknesses:**

I list the weaknesses and questions together here.

**[About the matching]**

I concur with the authors regarding the permutation invariant block in the EIT, acknowledging its potential to obviate the need for matching post-OCR. However, the rationale behind the decision to forego a straightforward matching step subsequent to OCR is not entirely clear to me. Is this choice motivated by a desire for increased flexibility, or are there other factors at play? From my perspective, matching algorithms can serve as modular, plug-and-play components, exemplified by their seamless integration in slot attention mechanisms as outlined in [1]. I recommend a more thorough elucidation of this particular point in the rebuttal, as it would greatly enhance the clarity and comprehensiveness of the explanation.

**[About the evaluation]**

In order to rigorously assess the contribution of each individual component within the algorithm’s design, I recommend broadening the scope of the ablation studies conducted. Specific areas to consider include: (1) experimenting with alternative OCR methodologies in lieu of DLP, to evaluate the framework’s adaptability and performance consistency across varying OCR techniques; (2) a detailed evaluation of the impact that each component recognized by DLP has on the ultimate policy learning. This is particularly pertinent for elements that do not share a direct correlation with dynamics and rewards, such as background features.

 **[About the compositional generalization]**

-  Can the method generalize to the case where the novel objects (e.g., different shape but similar to the ones seen in the training, e.g., cuboid versus cube) exist during the inference phase?

- Does the model possess capabilities for both extrapolation and interpolation with respect to the quantity of objects involved? To illustrate, consider a scenario wherein the model is trained on sets of 2, 4, 6, and 8 objects, and subsequently tested on sets of 3, 5, 7 (interpolation) as well as 1, 9, 10 (extrapolation). While I acknowledge the presence of some relevant results in Figure 5, a more systematic and thorough analysis of the model’s extrapolation and interpolation capabilities would be beneficial. This approach would align with the high-level conceptualization of generalization discussed in [2].

 **[About the interaction]**

-  I would appreciate additional clarification from the authors regarding the nature of entity interactions within the model. From my perspective, these interactions can be broadly classified into two categories: (1) interactions that influence dynamics without impacting the reward, and (2) interactions that affect both dynamics and reward. While I understand that the transformer is capable of capturing both types of interactions, a more explicit discussion on how it accomplishes this, and the implications of these interactions on the model’s performance, would be highly beneficial and contribute to a more thorough understanding of the model’s capabilities.

- I am interested in understanding how the density and frequency of interactions influence the performance of policy learning within the model. Could the authors possibly quantify and assess the model’s precision in predicting interactions across varying levels of interaction density and frequency? One potential metric for this evaluation could be the accuracy of the predicted entity state in comparison to the ground truth state, especially if direct interaction capture proves challenging within the simulator. I hypothesize that a reduction in workspace size, given a constant number of objects, is likely to increase interaction occurrences. Focusing on this aspect would provide valuable insights into the model’s robustness and adaptability under different operational conditions.

 **[About the presentation]**

Minor: I would recommend transferring the contents of either Appendix A or E to the main paper. This adjustment not only enhances the overall presentation but also efficiently utilizes the remaining available space (currently less than 9 pages).





*References*

[1] Locatello, Francesco, et al. "Object-centric learning with slot attention." Advances in Neural Information Processing Systems 33 (2020): 11525-11538.

[2] Balestriero, Randall, Jerome Pesenti, and Yann LeCun. "Learning in high dimension always amounts to extrapolation." arXiv preprint arXiv:2110.09485 (2021).

**Questions:**

I list the weaknesses and questions together in the above section.

---

> ### Author Response · Authors · 2023-11-19
>
> Thank you for the insightful and constructive feedback, we appreciate your acknowledgement of our work’s contribution.
>
> **Matching**
>
> The tasks we consider in our experiments require matching between entities in different views of the state as well as matching between state and goal entities. There are several reasons for our choice to take a learning approach for the matching rather than explicitly incorporating modular matching components in the algorithm. The overall rationale behind the decision is achieving flexibility, simplicity and robustness, as follows:
>
> - When working from image observations, a match for each entity does not always exist, for example in the case of occlusion in one of the viewpoints (happens frequently due to the robot arm). This case requires special handling in an explicit matching algorithm and there are not always a set of clear rules as to how to handle these discrepancies. This also requires incorporating additional hyper-parameters such as a threshold for what is considered a match in order to prevent matching with the next closest match even if it is not the same object (e.g matching a red cube to a purple cube by mistake). This increases complexity and hurts flexibility, as the hyper-parameters are not input dependent and may increase the chance of mismatches.
>
>
> - DLP often assigns multiple particles to a single object which creates a situation where there is not a one-to-one match. This makes it difficult to aggregate matching particles to a single entity of fixed size. Our learning approach deals with this automatically via the attention mechanism.
>
> - Generality: we designed our algorithm to have specific structure but try to keep it as general as possible. One direction for future work includes goal specification via natural language. In this case, the state tokens would be particles while the goal tokens would consist of natural language tokens. This is another case where the matching is not straightforward and a one to one match does not necessarily exist.
>
> In addition, explicit matching is not always a simple task. Matching entities in consecutive observations within a single-view object-centric framework poses significant challenges, as demonstrated in prior works (https://arxiv.org/abs/2107.09240). While incorporating matching algorithms, such as the Hungarian algorithm, is feasible, it introduces computational complexities that hinder scalability to datasets involving more than 3 objects. In our multi-view setting, this challenge intensifies as we must match entities across views, contending with potential occlusions and the absence of a one-to-one correspondence. Furthermore, the extensive literature on multi-view image matching (https://openaccess.thecvf.com/content_ICCV_2017/papers/Maset_Practical_and_Efficient_ICCV_2017_paper.pdf, https://arxiv.org/pdf/2205.01694.pdf) highlights the inherent difficulty of this problem, independent of the additional complexities associated with training a generative object-centric model.
>
> **Evaluation**
>
> (1) We agree this would be an interesting ablation. We intend to experiment with the slot-based Slot-Attention ([1] Locatello et al.), which was shown to perform well in the investigation performed in OCRL ([2] Yoon et al.).
> This ablation will require some time to find the right hyper-parameters for SLATE and integrate it with our code. We have started implementing this and will conclude the experiments after the rebuttal period.
>
> We pre-trained Slot-Attention (SA) using the same data as DLP with 10 slots (the maximum number feasible for 128x128 resolution given our computational resources). Notably, the training process was considerably slower than training DLP on similar hardware. Visual results can be found in the following image:  https://i.postimg.cc/PtWm4r1q/slots-25.png. It's important to highlight that, in the SA representation for our data, the robot and the table are often contained in the same slot, and occasionally, multiple cubes are grouped in a single slot instead of being individually represented. We are currently training our model using object-centric representations from the pre-trained Slot-Attention model. However, we anticipate that the imperfect SA decomposition may impact final performance, and we will provide detailed results in the upcoming version.
>
> Slot-Attention image columns: (1) original image, (2) reconstruction, (3) mask, (4) refined mask, (5) ->(14) slots.
>
>
>
> (2) We would like to emphasize that we discard the background features when extracting the latent representation from images, as it is constant in all of our experiments. We can add experiments that ablate some DLP components. Note that our new results on Push-T show that the policy makes use of the "feature" component in DLP, as it is the only feature that (implicitly) contains orientation information.

---

> > ### Author Response · Authors · 2023-11-19
> >
> > **Compositional Generalization**
> >
> > *Novel Objects* - This is an interesting question. Our method in its current version is not specifically designed to deal with novel objects zero-shot. Dealing with novel objects would require generalization from both the DLP and the EIT.
> > We would expect our method to zero-shot generalize to novel objects in case:
> > - They are visually similar to objects seen during training.
> > - Their physical dynamics are similar to the objects seen during training.
> >
> > To test our hypothesis, we deploy our trained agent in environments including modifications to shape, color and mass. Results are presented in Table 6 and discussed in Appendix B.2  in the new revision. Our main conclusion from this study is that while zero-shot generalization to objects with different properties is only partial, the success rates are not negligible and therefore hint at potential for few-shot generalization.
> >
> > *Extrapolation / Interpolation* - Due to long training times, we trained our image based method on at most 3 cubes. To test the interpolation capabilities of our method we trained an agent on a mixture of episodes from the 1-Cube and 3-Cubes environments and tested the performance on the 2-Cubes environment. The interpolation results for 2 cubes were similar to the results for 2 cubes with an agent trained only on 3 cubes.
> > For further insight on the extrapolation capabilities, we present a comparison of the performance of our agent with respect to the number of objects it was trained on for 1, 2 and 3 objects. Results are presented in Table 5 and discussed in Appendix B.2 in the new revision.
> >
> > **Interaction**
> >
> > We do not understand the questions regarding interaction. Since our method is model-free, the EIT is not trained to predict the next state but only the action (conditioned on the current observation and goal).
> > Therefore, we do not understand how to classify the interactions to dynamics/reward, nor how to report the precision in predicting interactions. We can visualize the attention pattern in the transformer, but it's not clear to us how to use this signal to answer the questions the reviewer raised. If we misunderstood the reviewer's question, we would be happy for a clarification.
> >
> > **Presentation**
> >
> > Thank you for the suggestion, we have moved additional details about the Chamfer reward to the main text.
> >
> > **References**
> >
> > [1] Locatello, Francesco, et al. "Object-centric learning with slot attention." Advances in Neural Information Processing Systems 33 (2020): 11525-11538.
> >
> > [2] Jaesik Yoon, Yi-Fu Wu, Heechul Bae, and Sungjin Ahn. An investigation into pre-training object-centric representations for reinforcement learning. arXiv preprint arXiv:2302.04419, 2023.

---

> > > ### Comment · Reviewer_sQ6n · 2023-11-20
> > >
> > > Thanks for the detailed feedback. Most of my concerns have been addressed. It would be nice to see the full ablation in the final or updated version in the future (as the authors promised). Additionally, the updated related work as well as the author's feedback on reviewer SBTK makes the work more complete.
> > >
> > > I would like to clarify the question I raised regarding interaction, I mean that sometimes the interactions might directly affect the reward/target. Like the case in this paper: https://openreview.net/forum?id=dYjH8Nv81K. However, the case in the paper I pointed out is not that relevant to those environments you considered in this work. So after reading the rebuttal and the current revision, I do not think it is a question for now.
> > >
> > > Given the detailed clarification and the technical contribution, I would keep the score.

---

### Author Response · Authors · 2023-11-19
**General Comments and Additional Experiments**

We would like to thank the reviewers for their detailed, constructive and insightful feedback.

We have responded to each reviewer individually. In addition, we provide this general response to address points that were brought up by multiple reviewers, as well as provide details about additional experiments that shed light on some of your questions/concerns. All changes and additions are highlighted in blue in the new revision (minor fixes not highlighted).

We detail the additions/modifications here:

1. We have added details about the **Chamfer Reward** to the main text.

2. We thank the reviewers for pointing out relevant papers to improve the scope of our **related work**. We have added suggested papers to the related work section, including several object-centric model-based RL, model-based planning and active inference algorithms. We explain in detail why SMORL is the most relevant baseline to our work.

3. We present preliminary experimental results on a new **Push-T** task in Appendix B.4 (video demonstrations in "Push-T" tab of our website). In this task, the agent must push a T-shaped block to a goal orientation specified by an image. We show results on $1$ and $2$ objects. The success in this task demonstrates the following additional capabilities of our proposed method:
- Handling objects that are not point-mass, which have more complex physical properties that affect dynamics.
- The ability of the EIT to infer object **properties that are not explicit** in the latent representation (i.e. inferring orientation from object latent visual attributes), and accurately manipulate them in order to achieve desired goals.

4. We widen the scope of our **zero-shot generalization** study in Appendix B.2 to address the following:

- Generalization to **unseen object properties**.
- Handling objects that are not small cubes.

To this end, we study zero-shot generalization to the following modifications of the objects:

- Cuboids obtained by enlarging either the x dimension or both x and y dimensions of the cubes.
- Star shaped objects with the same effective radius of the cubes seen during training.
- Cubes with different masses than in training.
- Cubes in colors not seen in RL training.
- Cubes in colors not seen in RL training nor DLP training.

The results indicate that our method facilitates few-shot generalization to changes in object properties (yet it handles some changes better than others, which is expected). More details and discussion can be found in Appendix B.2 of the new revision.

---

### Meta-Review · Area_Chair_qPTq · 2023-12-09

**Metareview:**

The paper addresses the important problem of object manipulation in robotics using learned policies from pixel observations. It proposes an approach that in nature is object centric -- the model can utilize a structured, object-centric representation of the environment. In particular, it consists of two main parts. First, a entity-centric representation of each image is learned (called particles) that are intended to correspond to objects. Second, a transformer is used as a actor and critic over these entities, that allows for more explicitly reasoning over individual entities and their relationships.

The paper has multiple exciting contributions. For one, it bakes in an important and simple inductive bias in often very generic neural networks -- the presence of entities in object manipulation. This, as the explanation and experiments demonstrates, leads to compositional generalization, e.g. different number of object at test or different colors at test than at train. It is refreshing to see that important challenges of a particular domain are addressed in a principle way.

Additionally, the approach introduces a reward as a distance between the current and goal states in the space of entities. Thus, the reward is more semantic in nature.

Finally, the approach is convincingly justified on a table-top manipulation setup from pixels.

Some of the concerns the reviewers bring up are some of the assumptions the work makes such as no partial observability that make it not fully ready for practical impact. Further, some reviewers want to see more comparisons against other works or evaluation on more than one environment, which is needed for the approach to be fully convincing.

**Justification For Why Not Higher Score:**

The paper would benefit from more thorough evaluation on larger set of environments. Further, algorithmically some of the assumptions such as no partial observability are to be addressed.

**Justification For Why Not Lower Score:**

The reviews are 3 x accept and 1 x borderline accept. Although there are some reservations w.r.t limited evaluations, all reviewers are excited by the ideas presented in the paper w.r.t. clever and simple integration of the notion of objects in the model and the reward, and the resulting benefits in terms of generalization. Hence, the paper is accepted to ICLR 2024.

---

### Decision · Program_Chairs · 2024-01-16

Accept (spotlight)